# An endothelial activin A-bone morphogenetic protein receptor type 2 link is overdriven in pulmonary hypertension

Gusty R. T. Ryanto [1,2], Koji Ikeda [1,3,4✉], Kazuya Miyagawa[1], Ly Tu [5,6], Christophe Guignabert [5,6], Marc Humbert[5,6,7], Tomoyuki Fujiyama [8], Masashi Yanagisawa [8], Ken-ichi Hirata[2] & Noriaki Emoto [1,2]

Pulmonary arterial hypertension is a progressive fatal disease that is characterized by pathological pulmonary artery remodeling, in which endothelial cell dysfunction is critically involved. We herein describe a previously unknown role of endothelial angiocrine in pulmonary hypertension. By searching for genes highly expressed in lung microvascular endothelial cells, we identify inhibin-β-A as an angiocrine factor produced by pulmonary capillaries. We find that excess production of inhibin-β-A by endothelial cells impairs the endothelial function in an autocrine manner by functioning as activin-A. Mechanistically, activin-A induces bone morphogenetic protein receptor type 2 internalization and targeting to lysosomes for degradation, resulting in the signal deficiency in endothelial cells. Of note, endothelial cells isolated from the lung of patients with idiopathic pulmonary arterial hypertension show higher inhibin-β-A expression and produce more activin-A compared to endothelial cells isolated from the lung of normal control subjects. When endothelial activin-A-bone morphogenetic protein receptor type 2 link is overdriven in mice, hypoxia-induced pulmonary hypertension was exacerbated, whereas conditional knockout of inhibin-β-A in endothelial cells prevents the progression of pulmonary hypertension. These data collectively indicate a critical role for the dysregulated endothelial activin-A-bone morphogenetic protein receptor type 2 link in the progression of pulmonary hypertension, and thus endothelial inhibin-β-A/activin-A might be a potential pharmacotherapeutic target for the treatment of pulmonary arterial hypertension.

[1] Laboratory of Clinical Pharmaceutical Science, Kobe Pharmaceutical University, Higashinada, Kobe, Japan. [2] Division of Cardiovascular Medicine, Department of Internal Medicine, Kobe University Graduate School of Medicine, Chuo, Kobe, Japan. [3] Department of Epidemiology for Longevity and Regional Health, Kyoto Prefectural University of Medicine, Kamigyou, Kyoto, Japan. [4] Department of Cardiovascular Medicine, Kyoto Prefectural University of Medicine, Kamigyou, Kyoto, Japan. [5] INSERM UMR_S 999, Le Plessis-Robinson, France. [6] Université Paris-Saclay, Université Paris-Sud, Le Kremlin-Bicêtre, France. [7] AP-HP, Service de Pneumologie, Centre de Référence de l'Hypertension Pulmonaire Sévère, DHU Thorax Innovation, Hôpital Bicêtre, Le Kremlin-Bicêtre, France. [8] International Institute for Integrative Sleep Medicine (WPI-IIIS), University of Tsukuba, Tsukuba, Japan. ✉email: ikedak@koto.kpu-m.ac.jp

Pulmonary arterial hypertension (PAH) is a rare but fatal disease, with an estimate of 15–50 cases per one million people worldwide[1–3]. Persistent pulmonary hypertension leads to right ventricular hypertrophy and ultimately results in right heart failure[4]. Because of the recent lowering of the diagnostic threshold for the mean pulmonary artery pressure from ≥25 to >20 mmHg, it is estimated that more people are at risk of having this serious condition than ever before[5,6]. Coupled with the fact that the current treatment options to halt the progression of this disease are still limited, the need to develop novel therapeutic strategies is increasing considerably[7]. The pathological hallmark of PAH is the intense structural and functional remodeling of pulmonary arteries in the lung, especially in precapillary artery region[8,9]. The detailed molecular mechanisms underlying vascular remodeling in PAH remain to be elucidated; however, endothelial cell (EC) dysfunction such as disordered angiogenesis with an inappropriate increase in proliferation and apoptosis has been revealed to be a major cause of this phenomenon[10–12].

Among the molecules that play a role in the development of PAH, bone morphogenetic protein receptor type II (BMPRII), a type-II receptor of the TGF-β superfamily, is the most studied and well-established key player in both clinical and laboratory settings[13–16]. This is because all forms of BMPRII mutations account for ~20% of idiopathic PAH (iPAH), whereas the number jumps to almost 80% in heritable PAH[14,17]. Moreover, patients with PAH with a BMPRII mutation show advanced disease progression, bad response to treatments, and increased morbidity and mortality rates[18]. Even in PAH without BMPRII mutations, the downregulation of BMPRII expression has been reported[15]. In addition, studies using ECs, blood outgrowth ECs, and induced pluripotent-stem-cell derived ECs from patients with PAH consistently showed that disruption of BMPRII signaling leads to the aforementioned EC dysfunction found in PAH[19–21]. All of these findings collectively underline the importance of BMPRII signaling in ECs in maintaining pulmonary vasculature physiology.

Ligands of TGF-β superfamily elicit cellular responses via seven type-I and five type-II receptors[22,23]. Ligand-binding leads to phosphorylation of the type-I receptor by the type-II receptor, and this activated heterodimeric type-I and type-II receptor complex triggers intracellular signaling through SMADs[22,23]. Despite the critical role of BMPRII in the pathology of PAH, relevance of the type-I receptors to PAH is less common. BMPRII forms heteromeric receptor complex with activin-like kinase type-1 (ALK1) in endothelial cells, while it mediates BMP signaling with ALK3 in smooth muscle cells[24]. Mutations in ALK1 cause hereditary hemorrhagic telangiectasia and rarely lead directly to the development of PAH[25]. It has also been reported that BMPRII signal deficiency causes pulmonary smooth muscle cell proliferation via alternative activation of ALK2/activin receptor type-2A[24].

ECs produce various soluble factors that play crucial roles in preserving homeostasis not only for blood vessels but also for multiple organs by communicating with organ-specific cells; however, a role of endothelial angiocrine in the pathogenesis of PAH remains to be unclear.

In this work, we describe a crucial role for pulmonary microvascular angiocrine in the maintenance of pulmonary artery homeostasis, and demonstrate that its dysregulated activation exacerbates pulmonary hypertension.

## Results

### INHBA is preferentially expressed in ECs isolated from the human lung microvasculature.
Because ECs play a central role in vascular remodeling associated with PAH and much of this remodeling occurs in precapillary pulmonary arteries, we focused on lung microvascular ECs-mediated angiocrine to reveal the unknown mechanisms underlying PAH. We searched for genes highly and preferentially expressed in lung microvascular ECs (hMVECs-L) via DNA microarray analysis using RNAs prepared from human ECs that were isolated from various vascular beds and RNAs prepared from various human tissues. Accordingly, we identified one gene that showed markedly high expression in hMVECs-L and in the lung compared to ECs isolated from other vascular beds and other tissues, respectively (Fig. 1a, b). This gene was inhibin-β-a (INHBA), which encodes the protein involved in the production of activin-A (ActA) that is a member of TGF-β superfamily. We confirmed the high expression levels of INHBA in hMVECs-L in comparison to ECs and smooth muscle cells (SMCs) isolated from various vascular beds using quantitative PCR (Fig. 1c). Of note, INHB-b and INH-α did not show such high expression in hMVECs-L (Fig. 1d). These data suggest a unique role for INHBA/ActA in the regulation of the pulmonary capillaries biology.

**INHBA/ActA negatively regulates the EC function**. In order to explore the role of INHBA/ActA in ECs, we overexpressed INHBA in pulmonary artery ECs (PAECs) and analyzed their angiogenic capacities. Overexpression of INHBA impaired tube formation on Matrigel and deteriorated the apoptosis induced by serum and growth factor-depletion in ECs (Fig. 2a and Supplementary Fig. 1a). Treatment with an endogenous ActA antagonist (i.e., follistatin) abolished the inhibitory effect of the INHBA overexpression on endothelial functions in PAECs (Fig. 2a and Supplementary Fig. 1a). In order to confirm the autocrine effect of ActA on EC, we prepared a conditioned medium of INHBA-overexpressing PAECs, which was enriched with INHBA/ActA (Supplementary Fig. 2e). Treatment with this conditioned medium inhibited tube formation and exacerbated apoptosis in PAECs (Fig. 2b and Supplementary Fig. 1a). The supplementation of follistatin abrogated the inhibitory effect of the conditioned medium on the EC functions (Fig. 2b and Supplementary Fig. 1a). In addition, recombinant ActA also showed an inhibitory effect on the EC functions, which was canceled by follistatin (Fig. 2c and Supplementary Fig. 1a). Follistatin treatment did not show significant effects on the EC functions in the control PAECs (Fig. 1a–c).

These data collectively suggest that overabundance of EC-derived INHBA/ActA could negatively regulate the EC function in an autocrine manner, though biological effects of overproduction of endogenously expressed INHBA/ActA need to be analyzed.

**Overproduction of INHBA/ActA reduces BMPRII protein levels in ECs**. Because ActA has an ability to bind to BMPRII, and BMPRII signaling is essential for maintaining the EC function, particularly in the pulmonary vasculature, we investigated BMPRII signaling as a possible intermediary mechanism for INHBA/ActA-mediated EC dysfunction. The overexpression of INHBA reduced the BMPRII protein expression in PAECs (Fig. 3a), while minimal changes in the BMPRII mRNA expression levels were detected (Supplementary Fig. 1d). The phosphorylation of SMAD1/5 was also reduced, and this was accompanied by decreased Id1 transcription in the PAECs that overexpressed INHBA (Fig. 3b, c). Similar to INHBA overexpression, treatment with recombinant ActA caused a reduction in the BMPRII protein levels and impaired its downstream signaling pathways in PAECs (Fig. 3d–f). We used GFP-overexpressing cells as a control for INHBA-overexpression cells; therefore, we assessed possible effects of the GFP-overexpression on the EC functions.

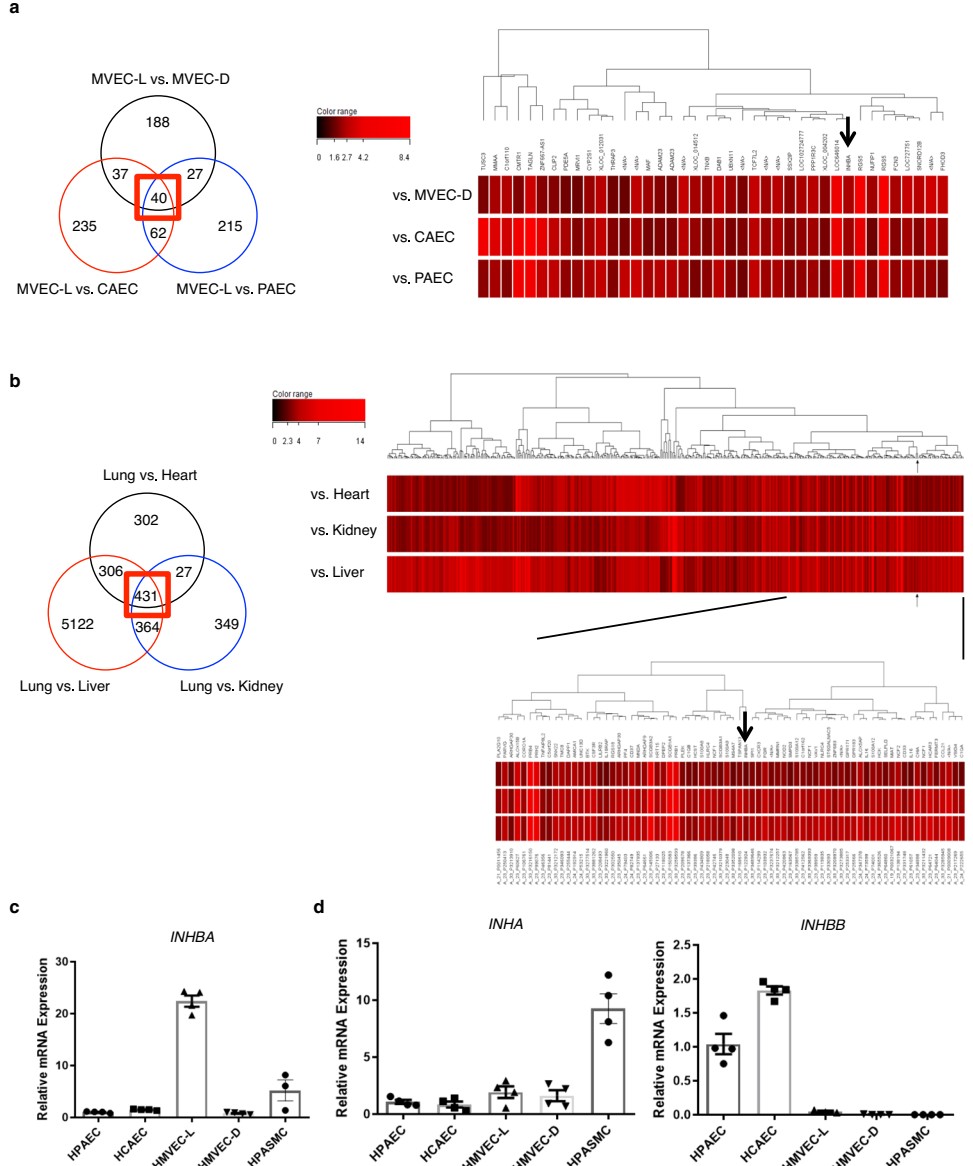

**Fig. 1 Identification of inhibin-β-A as a gene highly expressed in lung microvasculature. a** Genes whose expression in MVECs-L was three times greater than in MVECs-D, CAECs, or PAECs were separately identified through the DNA microarray analysis. Venn diagram analysis of these genes was shown. Numbers indicate the number of genes included in the sets of the intersections. We focused on genes whose expression in MVECs-L was three times greater than in all of MVEC-Ds, CAECs, and PAECs (the center intersection surrounded by red box). The heat map of these particular genes was shown. Arrow indicates INHBA. **b** Genes whose expression in the lung was five times greater than in the heart, kidney, or liver were separately identified through the DNA microarray analysis. Venn diagram analysis of these genes was shown. Numbers indicate the number of genes included in the sets of the intersections. We focused on genes whose expression in the lung was five times greater than in all of the heart, kidney, and liver (the center intersection surrounded by red box). The heat map of these particular genes was shown. Arrow indicates INHBA. **c, d** Quantitative PCR analysis for *inhibin-β-A* (*INHBA*) (**b**), and *inhibin-β-B* (*INHBB*), and *inhibin-α* (*INHA*) (**c**) in ECs and SMCs isolated from various vascular beds (n = 4 biologically independent cells in each group). HPAEC human pulmonary artery EC, HCAEC human coronary artery EC, HMVEL-D human microvascular EC in dermis, HPASMC human pulmonary artery SMC. Data are presented as the mean ± SEM.

Accordingly, we found no biological effects of GFP-overexpression on the functions and BMPRII protein expression in ECs (Supplementary Fig. 2a).

Because BMPRII continuously undergoes ligand-dependent internalization and subsequent recycling to the cell surface or degradation[16,26], we explored whether BMPRII downregulation was also induced by other ligands. In contrast to ActA, BMP-4, which is a canonical ligand for BMPRII, did not induce a reduction in the BMPRII protein levels in PAECs (Fig. 3g). Treatment with both ActA and BMP-4 caused robust phosphorylation of SMADs in PAECs (Supplementary Fig. 2b, c),

indicating that PAECs are sensitive to these ligands, whereas only ActA caused reduction of the BMPRII protein levels. We also treated vascular SMCs (VSMCs) with recombinant ActA, and found no effects on the BMPRII protein levels, while the robust SMAD2/3 activation was detected (Supplementary Fig. 2d).

In order to explore whether INHBA/ActA impairs the EC function through the BMPRII downregulation, we overexpressed BMPRII in PAECs with retrovirus-mediated gene transfection (Fig. 3h). This BMPRII overexpression affected neither tube formation nor apoptosis in control PAECs (Fig. 3i). In contrast,

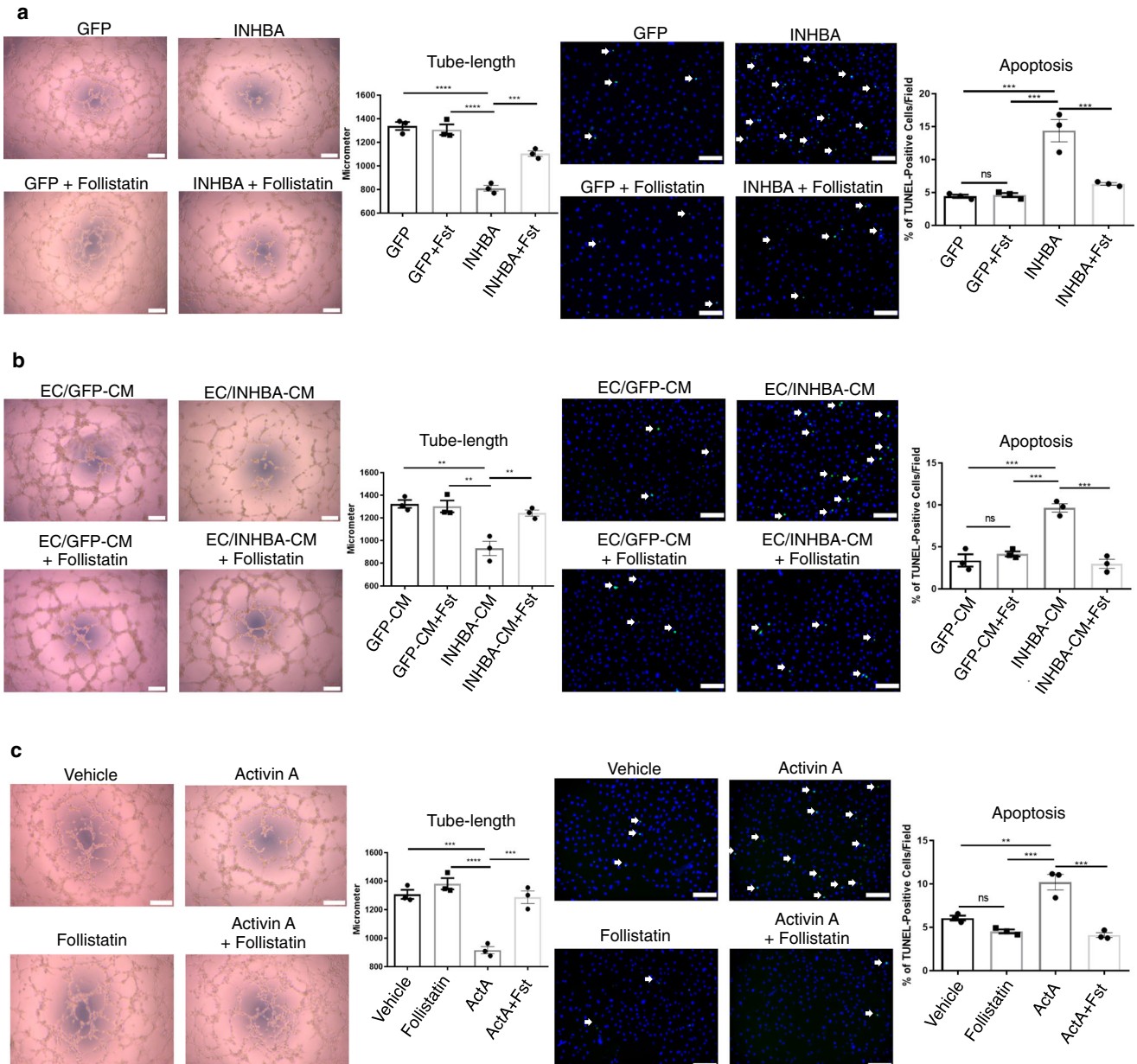

**Fig. 2 Excess INHBA/ActA-mediated angiocrine inhibits the angiogenic capacity in PAECs. a** Representative images and quantitation of tube length in a Matrigel tube-formation assay ($n = 3$ biologically independent values in each group) and apoptosis induced by serum starvation assessed by TUNEL staining ($n = 3$ biologically independent cells in each group) in PAECs transfected with either GFP or INHBA in the presence or absence of recombinant Follistatin (100 ng/mL). TUNEL-positive apoptotic cells are indicated by arrows. **b** Representative images and quantitation of a Matrigel tube-formation assay and apoptosis induced by $H_2O_2$ (200 μM/mL) in PAECs treated with conditioned medium (CM) derived from PAECs transfected with either GFP or INHBA in the presence or absence of recombinant Follistatin (100 ng/mL) ($n = 3$ biologically independent values in each group). TUNEL-positive apoptotic cells are indicated by arrows. **c** Representative images and quantitation of a Matrigel tube-formation assay ($n = 3$ biologically independent values in each group) and apoptosis ($n = 3$ biologically independent cells in each group) in PAECs treated with vehicle or ActA (20 ng/mL) in the presence or absence of recombinant Follistatin (100 ng/mL). TUNEL-positive apoptotic cells are indicated by arrows. Bars: 200 μm (tube-formation assays); 100 μm (apoptosis assays). $****P < 0.0001$; $***P < 0.001$; $**P < 0.01$; $*P < 0.05$. Exact $P$ values are shown in the Source data file. Data are presented as the mean ± SEM. One-way ANOVA with Tukey's post hoc test for multiple comparisons was used to compare the tube lengths and apoptotic cell counts between each group for all the figures.

BMPRII gene transfer improved the endothelial functions in PAECs that overexpress INHBA to levels similar to those of control cells (Fig. 3i). These data strongly suggest that INHBA/ActA inhibits the EC function largely by reducing BMPRII. We analyzed ActA concentration in the conditioned medium derived from these cells. BMPRII gene transfer did not affect the ActA production in either GFP-overexpressing or INHBA-overexpressing ECs (Supplementary Fig. 2e). Also, we assessed the expression of inflammatory cytokine, interferons, other members of INH family, other BMP type-I and type-II receptors, and Gremlin-1 in these cells. Neither INHBA nor BMPRII gene transfer affected the expression of most of these genes in PAECs, while interferon-β expression was enhanced by INHBA overexpression (Supplementary Fig. 2f).

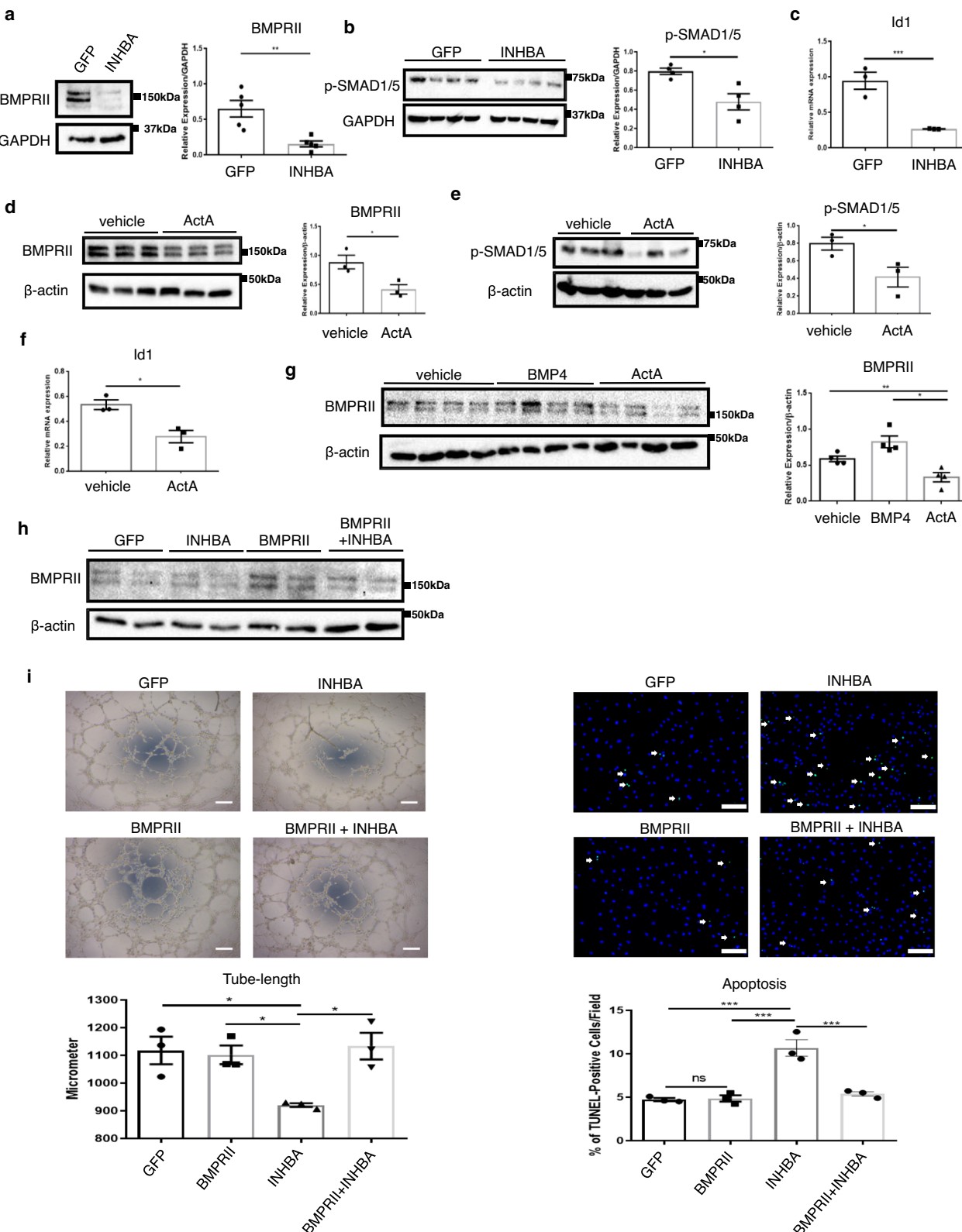

**ActA induces BMPRII lysosomal degradation in ECs**. Many receptors undergo ligand-induced internalization, mainly through clathrin-mediated endocytosis, and some of them are recycled back to the cell membrane, while others are targeted for degradation. Because INHBA/ActA reduced the BMPRII protein levels, despite the minimal changes in the mRNA expression, we presumed that INHBA/ActA might enhance BMPRII protein degradation. Using the cycloheximide-chase assay, we identified that ActA accelerated BMPRII protein degradation (Fig. 4a). In order to detect BMPRII trafficking, we prepared PAECs transfected with GFP-tagged BMPRII. BMPRII localized largely at the cell surface in PAECs (Fig. 4b and Supplementary Fig. 3a). When treated with ActA for 30 min, cell-surface-localized BMPRII disappeared and it accumulated in the intracellular organelles,

**Fig. 3 Overabundance of INHBA/ActA impairs the EC function by reducing BMPRII. a, b** Representative immunoblots and densitometric analysis for BMPRII (**a**) ($n = 5$ biologically independent cells in each group) and phospho-SMAD1/5 (**b**) ($n = 3$ in each group) in PAECs transfected with either GFP or INHBA. **c** Quantitative PCR for Id1 in PAECs transfected with either GFP or INHBA ($n = 3$ biologically independent cells in each group). **d, e** Representative immunoblots and densitometric analysis for BMPRII (**d**) ($n = 4$ in each group) and phospho-SMAD1/5 (**e**) ($n = 3$ biologically independent cells in each group) in PAECs treated with either vehicle or ActA (20 ng/mL) for 6 h. **f** Quantitative PCR for Id1 in PAECs treated with either vehicle or ActA ($n = 3$ biologically independent cells in each group). **g** Representative immunoblots and densitometric analysis of BMPRII in PAECs treated with either vehicle, BMP-4 (20 ng/mL), or ActA (20 ng/mL) for 6 h ($n = 4$ biologically independent cells in each group). **h** Immunoblotting for BMPRII in PAECs transfected with either GFP, INHBA, BMPRII, or both INHBA and BMPRII. Similar results were obtained in three independent experiments. **i** Matrigel tube-formation assay ($n = 3$ biologically independent values in each group) and TUNEL staining for apoptosis ($n = 3$ biologically independent cells in each group) in PAECs transfected with either GFP, INHBA, BMPRII, or both INHBA and BMPRII. Arrows indicate TUNEL-positive apoptotic cells. Bars: 200 μm (tube-formation assay); 100 μm (apoptosis assay). \*\*\*$P < 0.001$; \*\*$P < 0.01$; \*$P < 0.05$; #, not significant ($P > 0.05$). Exact $P$ values are shown in the Source data file. Data are presented as the mean ± SEM. Two-sided Student's $t$-test was used to analyze the difference between each study group in all of the western blot quantitation procedures (**a**, **b**, **d**, and **e**) and quantitative PCR (**c** and **f**). One-way ANOVA with Tukey's post hoc test for multiple comparison was used to analyze the differences between each study in all of the western blot quantitation procedures (**g**) and in tube lengths and apoptotic cell counts between each group (**i**).

where LysoTracker was detected (Fig. 4b). In contrast, a significant amount of BMPRII remained on the cell surface when treated with BMP-4 for 30 min, whereas some of the BMPRII appeared to be targeted to lysosomes (Fig. 4b). The colocalization of BMPRII and LysoTracker was quantified using Pearson's correlation coefficients. ActA mediated significant colocalization of BMPRII and Lysotracker in PAECs, while less colocalization of them was observed in cells treated with BMP-4 (Supplementary Fig. 3b).

In order to further validate the ActA-mediated internalization and targeting to lysosomal degradation, we treated the cells with PitStop, a clathrin-mediated endocytosis inhibitor, and bafilomycin A, a lysosome inhibitor, in addition to cycloheximide for 30 min. Both PitStop- and bafilomycin A abrogated the BMPRII reduction induced by ActA (Fig. 4c, d). To further confirm the lysosomal degradation, we treated the cells with leupeptin, a lysosomal protease inhibitor. Treatment with leupeptin abolished the BMPRII reduction induced by ActA in a way similar to bafilomycine A (Supplementary Fig. 3c). Consistently, PitStop treatment canceled the ActA-mediated BMPRII targeting to lysosomes, whereas bafilomycin treatment strengthened the BMPRII signals that accumulated in the lysosomes in cells treated with both ActA and BMP-4 (Fig. 4e, f).

We then examined whether ActA binding activates BMPRII as other canonical ligands do. Serine/threonine phosphorylation of BMPRII was induced in PAECs after treatment with ActA in a way similar to BMP-4 treatment, suggesting that ActA might not be a decoy ligand (Fig. 4g). However, all phosphorylated serines and threonines were detected in this assay; therefore, this result does not necessarily mean the phosphorylation events in BMPRII mediated by BMP-4 and ActA are identical, and it remains unclear whether ActA could induce intracellular signaling through BMPRII in ECs.

**Dysregulated activation of INHBA in ECs exacerbates pulmonary hypertension.** In order to determine whether dysregulated INHBA/ActA-mediated angiocrine affects the development of PAH, we generated mice in which INHBA was overexpressed under the control of the VE-cadherin promoter (VEcad-INHBA-Tg). VEcad-INHBA-Tg mice were viable and fertile, and there were no significant differences of body weight between littermate wild-type (WT) and the Tg mice both in neonates and adults (Supplementary Fig. 4a). We confirmed that INHBA is overexpressed in ECs isolated from the lungs of VEcad-INHBA-Tg mice (Supplementary Fig. 4b). Consistently, ActA concentration in the conditioned medium was significantly higher in lung ECs isolated from the Tg mice than in WT mice, whereas serum ActA levels were not different between these mice (Supplementary

Fig. 4c). We also confirmed that INHBA expression in bone marrow cells was similar between these mice (Supplementary Fig. 4d). There were no apparent morphological changes in the lung vasculatures, and lung vessel density was not different in neonates of the Tg mice comparing to those in WT mice (Supplementary Fig. 4e). Also, no significant cardiac defects were detected in the Tg mice (Supplementary Fig. 4f). We also analyzed the expression of other INH family members in the lungs and heart, and found no significant difference between these mice (Supplementary Fig. 4g). We then assessed changes in hemodynamics and the heart function in these mice. There were no differences in the blood pressure, heart rate, and heart left ventricular function between VEcad-INHBA-Tg and WT mice (Supplementary Fig. 5a, b). In contrast, a significant decrease in the pulmonary artery acceleration time (PAAT) was observed in VEcad-INHBA-Tg mice compared to WT mice (Supplementary Fig. 5b). Consistent with the decreased PAAT, right ventricular systolic pressure (RVSP) was significantly increased in VEcad-INHBA-Tg mice compared to WT mice, even under normoxic conditions (Fig. 5a). Right heart hypertrophy, assessed by the Fulton index, was modestly but significantly deteriorated in VEcad-TRF2DN-Tg mice under normoxic condition (Fig. 5b). Consistently, the number of muscularized PA increased, while the number of distal PA decreased in VEcad-INHBA-Tg mice under normoxic condition (Fig. 5c–e, Supplementary Fig. 6a). Of note, BMPRII and phosphorylated SMAD1/5 protein levels, and Id1 transcription levels were significantly reduced without changes in the BMPRII mRNA expression in the whole lungs of VEcad-INHBA-Tg mice compared to WT mice under normoxic condition (Fig. 5f–h, Supplementary Fig. 6b).

When pulmonary hypertension was induced by three weeks of exposure to 10% hypoxia, VEcad-INHBA-Tg mice exhibited exacerbated pulmonary hypertension, accompanied by deteriorated right ventricular hypertrophy compared to WT mice (Fig. 5a, b). Histological analysis revealed the destruction of the EC single-layer lining and a high level of muscularization in the PA of VEcad-INHBA-Tg mice, whereas the distal PA showed a decrease in the Tg mice (Fig. 5c–e, Supplementary Fig. 6b). BMPRII and phosphorylated SMAD1/5 protein levels, and Id1 transcription levels were markedly reduced without changes in the BMPRII mRNA expression in the whole lungs of VEcad-INHBA-Tg mice compared to WT mice (Fig. 5f–h, Supplementary Fig. 6a). These results indicate that the overproduction of INHBA in ECs exacerbates pulmonary hypertension in association with the BMPRII signal deficiency in the lungs.

**Conditional knockout of INHBA in ECs ameliorated pulmonary hypertension.** Because a null mutation of INHBA has

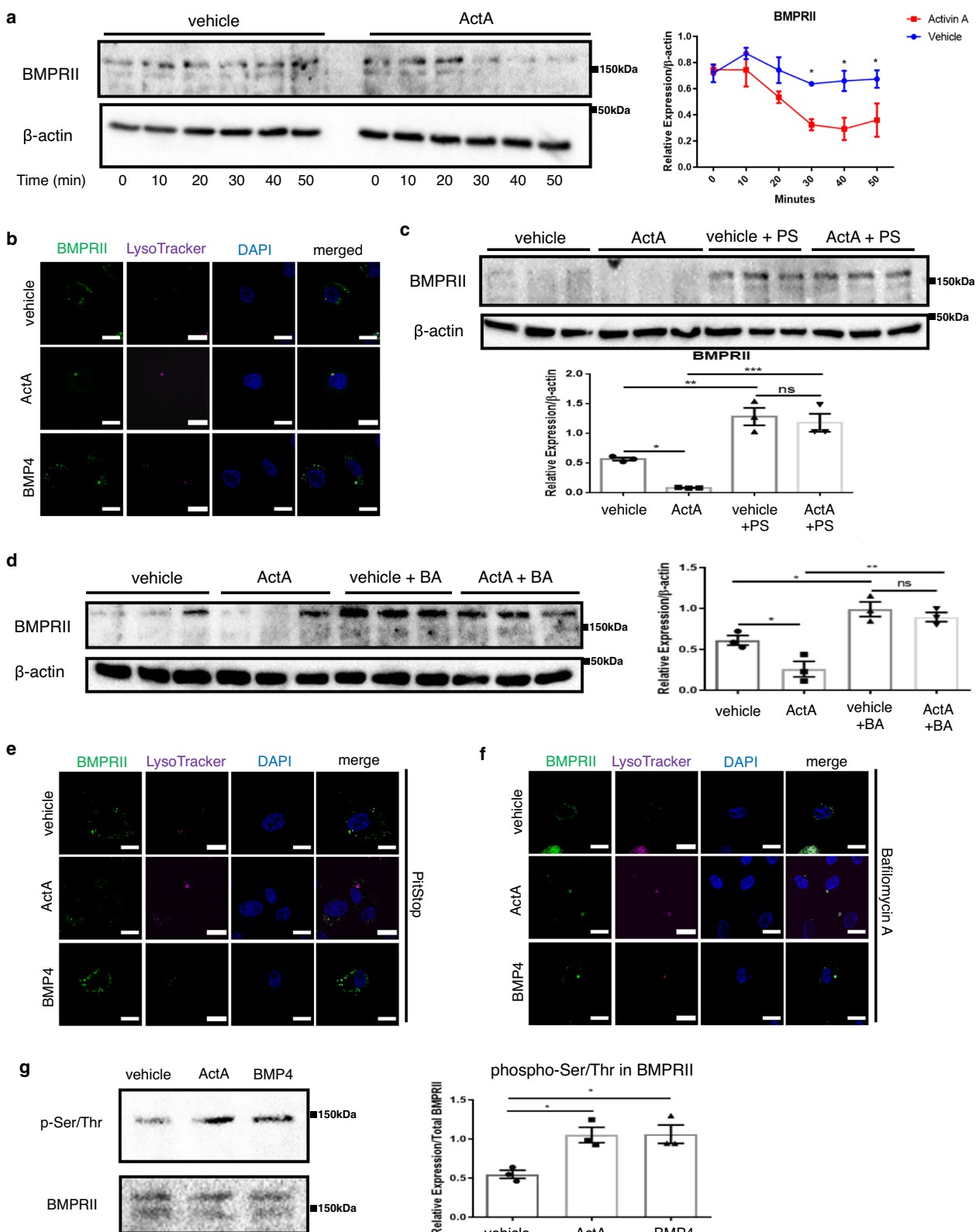

been reported to be lethal in mice, we generated EC-specific conditional INHBA knockout (INHBA-ECKO) mice by mating INHBA-flox and VEcad-Cre-Tg mice, to further analyze the role of INHBA/ActA-mediated angiocrine in PAH. INHBA-ECKO mice were viable and fertile, and there were no significant differences in body weight between INHBA-flox and the INHBA-ECKO mice both in neonates and adults (Supplementary Fig. 7a).

EC-specific deletion of INHBA was confirmed using ECs and non-ECs isolated from the lungs of INHBA-ECKO and INHBA-flox mice, while serum ActA levels were similar between these mice (Supplementary Fig. 7b, c). There were no apparent morphological changes in the lung vasculatures in neonates, and no significant cardiac defects were detected in the ECKO mice (Supplementary Fig. 7d, e). We also analyzed the expression of

**Fig. 4 ActA accelerates ligand-mediated BMPRII endocytosis and lysosomal degradation. a** Cycloheximide-chase assay for BMPRII in PAECs treated with either vehicle or ActA (20 ng/mL) ($n = 3$ biologically independent cells in each group). **b** Representative immunocytochemistry images for GFP-tagged BMPRII (in green color) in PAECs treated with either vehicle, BMP-4 (20 ng/mL), or ActA (20 ng/mL) for 30 min. Lysosomes were visualized using LysoTracker (in magenta color). Bars: 20 μm. Similar results were obtained in three independent experiments. **c**, **d** Immunoblots and densitometric analysis for BMPRII in PAECs pretreated with either PitStop (PS) (**c**) or bafilomycin A (BA) (**d**) for 2 h, followed by treatment with either vehicle or ActA for 30 min in the presence of cycloheximide ($n = 3$ biologically independent cells in each group). **e**, **f** Immunocytochemistry for GFP-tagged BMPRII (in green color) and lysosomes (in magenta color) in PAECs pretreated with either PitStop (**e**) or bafilomycin A (**f**), followed by treatment with either vehicle, BMP-4, or ActA. Bars: 20 μm. Similar results were obtained in four independent experiments. **g** Representative immunoblots and densitometric analysis for serine/threonine-phosphorylated BMPRII in PAECs treated with vehicle, BMP-4, or ActA ($n = 3$ biologically independent cells in each group). ***$P < 0.001$; **$P < 0.01$; *$P < 0.05$. Exact $P$ values are shown in the Source data file. Data are presented as the mean ± SEM. Two-sided Student's $t$-test was used to analyze the differences between the vehicle and ActA-treated groups at each time point (**a**). One-way ANOVA with Tukey's post hoc test for multiple comparisons was used to analyze the differences between each study group in all of the western blot quantitation procedures (**c**, **d**, and **g**).

other INH family members in the lungs and heart, and found no significant difference between these mice (Supplementary Fig. 7f). Under normoxic condition, the blood pressure, heart rate, pulmonary artery pressure, and the Fulton index were similar between the ECKO and the flox mice (Fig. 6a, b, Supplementary Fig. 8a). Consistently, no significant vascular remodeling was detected in the lungs of INHBA-ECKO mice under normoxic condition (Fig. 6c–e).

After 3 weeks of treatment with hypoxia, the heart function, systolic blood pressure, and PAAT assessed by echocardiography were not different between INHBA-ECKO and INHBA-flox mice (Supplementary Fig. 8b). Notably, pulmonary hypertension was ameliorated, accompanied by less right heart hypertrophy in INHBA-ECKO mice compared to INHBA-flox mice after chronic exposure to hypoxia (Fig. 6a, b). Consistently, vascular remodeling was ameliorated in INHBA-ECKO mice compared to that in INHBA-flox mice (Fig. 6c–e). These data further support the critical role of EC-derived INHBA/ActA in the progression of PAH.

**INHBA/ActA is overproduced in pulmonary ECs isolated from patients with idiopathic pulmonary hypertension.** In order to assess whether INHBA/ActA-mediated angiocrine is involved in the pathogenesis of PAH in clinical settings, the expression and production of INHBA/ActA were examined using ECs isolated from the lungs of patients with iPAH or normal control subjects. The INHBA mRNA expression levels were significantly higher in iPAH-ECs than in control ECs (Fig. 6f). Consistently, the production of ActA was also enhanced in iPAH-ECs compared to control ECs (Fig. 6g). Because hypoxia is an important modifier in the pathogenesis of PAH, we examined whether hypoxia affects the INHBA expression in these human ECs. After treatment with 1% hypoxia, the INHBA mRNA expression levels were enhanced in both iPAH-ECs and control ECs (Fig. 6h).

We then analyzed whether the conditioned medium prepared from patient-derived lung ECs affects the EC functions. Treatment with the conditioned medium derived from iPAH-ECs impaired tube formation in PAECs compared to that derived from control ECs (Fig. 6i). Of note, this negative effect of the conditioned medium from iPAH-ECs was canceled by the addition of follistatin (Fig. 6i). These data strongly suggest a crucial role for dysregulated INHBA/ActA-mediated angiocrine in the development of PAH in clinical settings.

## Discussion
In this study, we identified a role for dysregulated angiocrine in pulmonary capillaries in the progression of vascular remodeling and development of pulmonary hypertension. We identified INHBA/ActA as an angiocrine factor highly preferentially produced by lung microvascular ECs. INHBA/ActA negatively regulates the EC function in an autocrine manner by inducing

downregulation of BMPRII. Gain and loss of INHBA/ActA-mediated angiocrine in mice revealed that dysregulation of this unique angiocrine is critically involved in the pathogenesis of PAH. Moreover, INHBA/ActA overabundance was also replicated in a translational setting using ECs isolated from the lungs of patients with iPAH.

ActA, which is formed by the homodimerization of inhibin-β, is a member of the activin/inhibin group of the TGF-β superfamily[27]. Although it was first found in gonadal cells, recent findings showed that it is a multifunctional cytokine that affects angiogenesis, inflammation, proliferation, and apoptosis[27–30]. ActA binds and activates its own type-II receptor (activin receptor IIA or IIB), which then activates the type I receptor and subsequent SMAD pathways[27]. Interestingly, it has also been reported that ActA interacts with other members of the type-II TGF-β superfamily receptors, including BMPRII, although its physiological relevance remained unknown[29,31]. BMPRII downregulation has been observed in all PAH forms, and dysregulated BMPRII trafficking has been implicated in PAH development[16,21,26]. It has been reported that lysosomal degradation is a crucial pathway in BMPRII regulation and the inhibition of lysosomal degradation could lead to preserved BMPRII expression[32]. Internalization of BMPRII, which happens largely through clathrin-mediated endocytosis, is also deemed to be important in the regulation of BMPRII expression and activity[21,33,34]. Our study showed that INHBA/ActA reduces the BMPRII levels in ECs by accelerating its targeting to lysosomes for degradation after clathrin-mediated endocytosis. There are other internalization and degradation pathways, such as the caveolin-mediated endocytosis and proteasomal degradation of BMPRII, and the role for INHBA/ActA in these pathways remains to be examined[21,33–35]. Also, a non-specific action of PitStop in the inhibition of endocytosis has been reported, which requires caution to interpret the data[36]. In addition, we used exogenously expressed BMPRII-GFP to analyze the subcellular localization of BMPRII in PAECs. Careful interpretation is required because overexpressed proteins could show different behaviors from endogenous ones. Despite the critical role of BMPRII in PAH, it is well known that BMPRII mutation penetrance in patients with PAH is only ~20% in clinical settings. This suggests that other factors or molecules besides BMPRII mutation must be involved if an individual develops PAH[17,37]. Because INHBA/ActA is involved in the development of PAH largely through reducing BMPRII, it may play less role in the PAH pathology in the context of reduced BMPRII. However, overabundance of INHBA/ActA could mediate further reduction of BMPRII in ECs in addition to the BMPRII deficiency due to other causes; therefore, INHBA/ActA might be a possible missing piece in the molecular mechanisms underlying PAH[29].

ActA is thought to exert its biological functions through binding to and activating canonical activin receptors; however, its ability to bind to other receptors in the TGF-β superfamily has

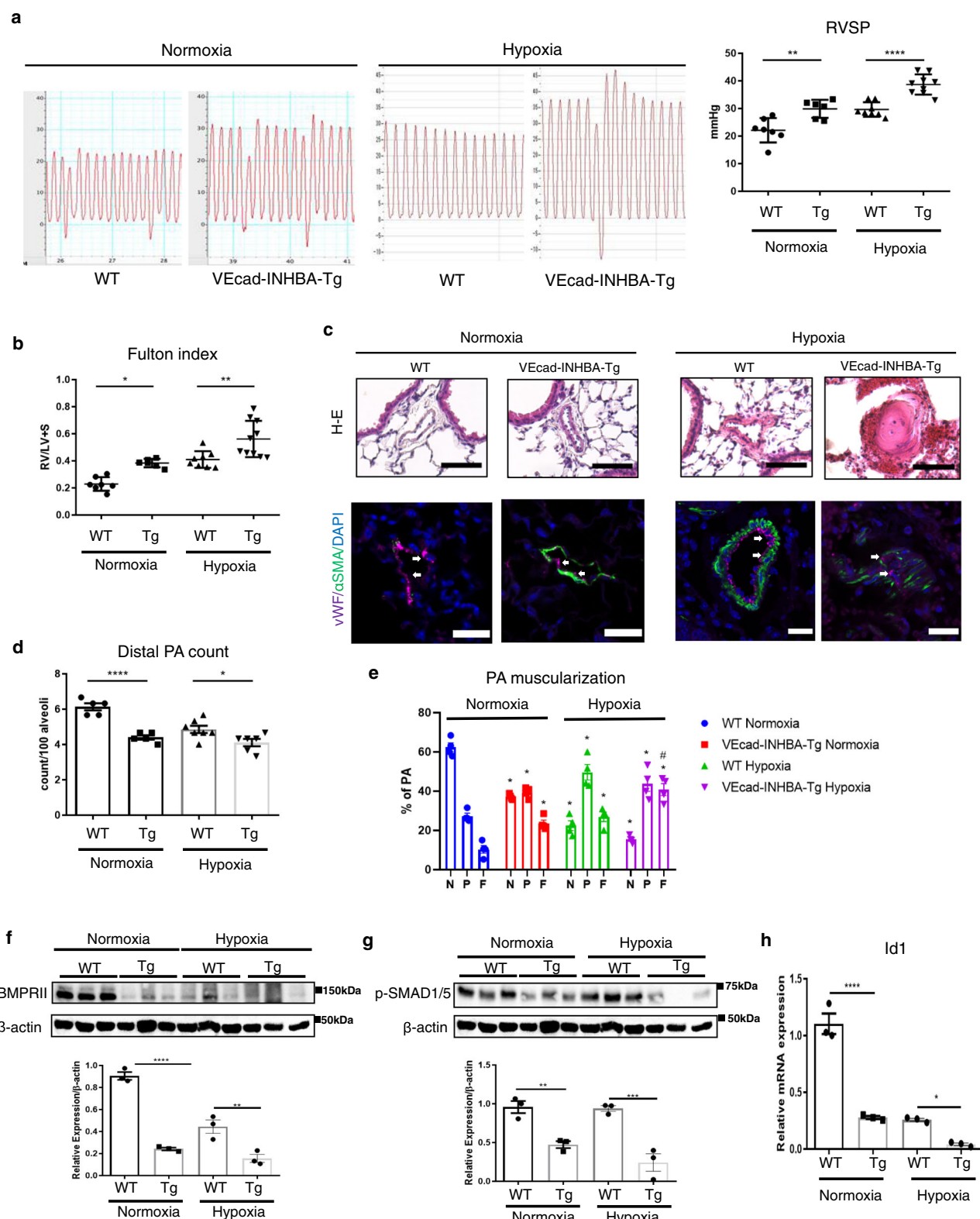

been reported[27,38]. Our results clearly showed that ActA mediates BMPRII internalization and subsequent lysosomal degradation, leading to the BMPRII signal deficiency in ECs. TGF-β super-family members, including BMPs and ActA, utilize type-I and II receptors to activate intracellular signaling pathways[39]. There are seven type-I receptors, and it remains unknown which type-I receptor is coupled to BMPRII in the event of ActA binding[40]. It is even possible that ActA-mediated BMPRII internalization occurs in the absence of coupling with type-I receptors. Sur-prisingly, ActA did not affect the BMPRII expression levels in VSMCs. Because the preferentially expressed type-I receptor(s) is different in different cell types, a type-I receptor(s) that is expressed in ECs but not in VSMCs might be needed for ActA to mediate BMPRII internalization.

**Fig. 5 Target activation of INHBA in ECs exacerbates PH in association with impaired BMPRII signaling. a** Representative right ventricle pulse waves and RVSP in WT and VEcad-INHBA-Tg mice under either normoxic or 3-week hypoxic (10% $O_2$) conditions ($n = 7$ biologically independent animals for normoxia WT; $n = 6$ for normoxia TG; $n = 7$ for hypoxia WT; $n = 9$ for hypoxia Tg). **b** Fulton's index (right ventricular to left ventricular plus septum weight ratio) measurements in WT and VEcad-INHBA-Tg mice ($n = 7$ biologically independent animals for normoxia WT; $n = 6$ for normoxia TG; $n = 8$ for hypoxia WT; $n = 10$ for hypoxia Tg). **c** Representative images of hematoxylin and eosin and immunofluorescence staining with an EC marker (vWF; in magenta color) and SMC marker (α-SMA; in green color) with DAPI in the lungs of WT and VEcad-INHBA-Tg mice. Arrows indicate vWF-positive ECs. Bars: 50 μm. Similar results were obtained in five biologically independent samples. **d, e** Quantitation of the distal pulmonary artery (20–50 μm) count per 100 alveoli (**d**) ($n = 5$ biologically independent values for normoxia WT and normoxia Tg; $n = 7$ biologically independent values for hypoxia WT; $n = 6$ biologically independent values for hypoxia Tg) and muscularized distal pulmonary artery (**e**) ($n = 4$ biologically independent values in each group) in the lungs of WT and VEcad-INHBA-Tg mice. The number of no-muscularized (N), partially muscularized (P), and fully muscularized distal arteries were counted. **f, g** Immunoblots and densitometric analysis for BMPRII (**f**) and phospho-SMAD1/5 (**g**) in the lungs of WT and VEcad-INHBA-Tg mice ($n = 3$ biologically independent samples in each group). **h** Quantitative PCR for Id1 in ECs isolated from the lungs of WT and VEcad-INHBA-Tg ($n = 3$ biologically independent samples in each group). ****$P < 0.0001$; **$P < 0.01$; *$P < 0.05$; #, not significant ($P > 0.05$). Exact $P$ values are shown in the Source data file. Data are presented as the mean ± SEM. Two-sided Student's $t$-test was used to analyze the differences between the WT and VEcad-INHBA-Tg groups in the distal PA count (**d**), BMPRII and pSMAD1/5 western blot quantitation (**f** and **g**), and Id1 mRNA expression levels (**h**). One-way ANOVA with Tukey's post hoc test for multiple comparisons was used to analyze the differences between each study group in the RVSP and Fulton index measurements (**a** and **b**). Two-way ANOVA with Tukey's post hoc test for multiple comparisons was used to analyze the differences between each study group in the PA muscularization (**e**).

---

It has been reported that ActA enhances the proliferation in VSMCs[41]. Because ActA is secreted extracellularly, it is plausible that EC-derived ActA could also affect other cells adjacent to ECs, most notably SMCs. Our data strongly suggest that the over-abundance of ActA in ECs exacerbates pulmonary hypertension by impairing the EC function due to BMPRII deficiency; however, other mechanisms, such as enhancing the SMC proliferation, could also be involved in the causal relationship between EC-derived ActA and PAH. Also, we found that INHBA overexpression increased interferon-β expression in PAECs. Considering a potential role of interferon in the development of PAH, INHBA-mediated overproduction of interferon-β may also be involved in the mechanism underlying the PAH associated with ActA overabundance[42].

Although it has been demonstrated that BMPRII activity preservation is important in preventing and reversing vascular remodeling in PAH, no treatment options to restore the BMPRII expression have been established clinically[14,43]. Our data suggest that ActA-mediated angiocrine might be a promising therapeutic target to restore the BMPRII function. In this regard, follistatin, an endogenous inhibitor for ActA is a good candidate for the treatment of PAH[44]. However, it antagonizes various TGF-β superfamily members, such as myostatin, and in fact, its clinical application has been intensively studied regarding Duchenne muscular dystrophy[45]. Developing a ligand trap (follistatin-based) specific for ActA would constitute a breakthrough in the treatment of PAH. In fact, Sotatercept, a ligand trap with high selectivity for multiple members of TGF-β superfamily including ActA, is now undergoing the phase 2 clinical trial in patients with PAH. Our study revealed that INHBA/ActA-mediated pulmonary microvascular angiocrine is a previously unknown modifier in the development PAH. Further clinical studies are warranted to validate INHBA/ActA as a therapeutic target in the treatment of PAH.

## Methods

**Reagents**. Reagents used for experiments are shown in Supplementary Table 3.

**DNA microarray**. Human PAECs and human SMCs (PASMCs) were purchased from PromoCell. Human microvascular ECs of the dermis and the lings (MVECs-D and MVECs-L) and human coronary artery ECs (CAEC) were purchased from Clonetics. PAECs and CAECs were cultured in HuMedia-EG2 (Kurabo), and hMVECs-L and hMVECs-D were cultured in EGM-MV2 (Lonza). When the cells became subconfluent (~90%), RNA was extracted from cultured ECs using TRIzol (Invitrogen). RNAs for 5-donor pooled human lungs, heart, kidneys, and liver obtained from BioChain Institute. Labeled cRNA was prepared using 1 vial of the mixed RNAs for each sample according to the One-Color Microarray-Based Gene Expression Analysis ver. 6.5 protocol, and 600 ng of cRNAs was used for hybridization with microarray chips (SurePrint G3 Human GE; G4851B) for 17 h. After washing, signals were scanned using an Agilent DNA microarray scanner, and the data were analyzed using Feature Extraction ver. 10.7.1.1 (Agilent Technologies).

**INHBA and BMPRII plasmid construction and retrovirus production**. Human INHBA DNA was prepared with RT-PCR using Pfx polymerase (Invitrogen), followed by subcloning into a pCR-Blunt II-TOPO vector (Invitrogen). The INHBA insert was cut out, and subcloned into a pMSCVneo vector (Clontech). In order to prepare the expression construct for GFP-tagged BMPRII, DNA for whole-length human BMPRII was prepared by RT-PCR using Phusion High-Fidelity PCR Master Mix with HF Buffer (New England Biolabs), followed by subcloning into a pCR-Blunt II-TOPO vector (Invitrogen). The BMPRII insert was then cut out, and subcloned into a pAcGFP1-N1 plasmid (Clontech). Finally, the GFP-BMPRII insert was cut out and subcloned into the pMSCVneo vector (Clontech).

The INHBA/pMSCVneo and BMPRII-GFP/pMSCVneo constructs, alongside the pMSCVneo/GFP control construct, were transfected into GP2-293 packaging cells (Clontech) using a Lipofectamine 3000 (Thermo-Fisher) concurrent with a pVSVG viral envelope expression construct, followed by changing the medium with a fresh growth medium in 24 h. After an additional 24 h, the medium was changed again with a fresh growth medium and incubated for another 24 h. Subsequently, the culture medium containing retroviruses was collected and stored at −80 °C after the removal of floating cells by centrifugation.

**Retroviral transfection in PAECs and conditioned medium production**. Frozen stocks of the viruses were thawed immediately before use. We did not strictly titrate the viruses; however, we regularly determined the appropriate dilution of retroviruses by analyzing the transduction efficacy of GFP-positive control whenever viruses were newly prepared. PAECs at ~70% confluence were infected with retroviruses using a 1:1-4 mixture of the retroviruses-containing media and growth medium in the presence of 8 μg/mL polybrene (Sigma-Aldrich). After 24 h, a fresh growth medium was added, and the cells were incubated for another 24 h prior to use in the experiments.

In order to prepare the conditioned medium, PAECs transfected with INHBA or GFP were cultured in a fresh growth medium for 24 h, followed by the collection of the culture medium. After centrifuged at $450 \times g$ for 5 min, the supernatants were stored at −80 °C until use in the experiments.

**Tube-formation assay**. Prior to seeding, 96-well plates were coated with 50 μL/well of Matrigel (Corning) and incubated at 37 °C for 30 min. PAECs were then seeded at 20,000 cells/well in the growth medium and further incubated at 37 °C for 6–8 h. Images were taken for each well every 1 h after 4 h using a Zeiss Primovert Inverted Microscope (Zeiss) attached to a Zeiss Axiocam ERc 5 s (Zeiss), to determine the optimal time point for the quantitative analysis. Images in low magnification (×4) were captured for each well. Each image was divided into 16 rectangles, and 10 rectangles were randomly chosen. The lengths of tubes between the branching junctions were measured and summed in each single rectangle, and then values obtained from 10 rectangles were averaged. A measurement example is shown in Supplementary Fig. 9. We performed experiments 3 independent times using 1–2 wells for each group.

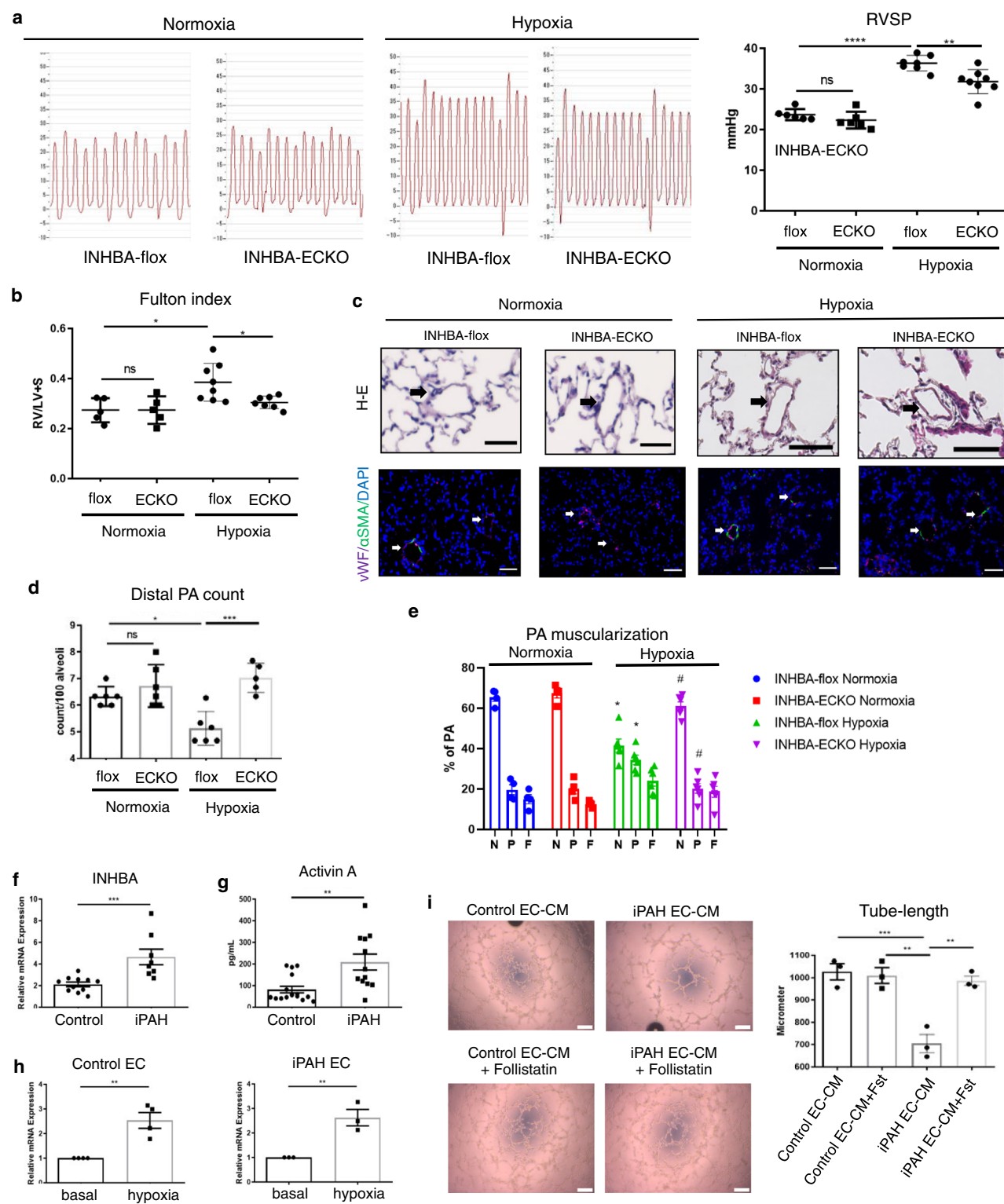

**Apoptosis assay**. In order to assess the PAEC apoptosis vulnerability, cells were seeded at 10,000 cells/well on 96-well plates and cultured overnight to allow adherence. On the next day, the growth medium was changed to a serum- and growth factor-free medium, and the cells were incubated for an additional 24 h. For the cells cultured in the conditioned medium containing serum and growth factors, apoptosis was induced by oxidative stress using 200 μM hydrogen peroxide. Subsequently, the cells were fixed with 4% PFA, followed by permeabilization with 0.2% Triton X-100 (Wako) in PBS, and then subjected to nuclear staining using Hoechst 33342. Cells with bright condensed chromatin signals were considered to be cells undergoing apoptosis. Apoptosis was assessed using TUNEL- and AnnexinV/PI-staining of the cells. The number of apoptotic cells was counted and divided by the total number of cells in three fields (×20 magnification) for each well. Images were obtained using a Keyence BZ-X800 fluorescence microscopy (Keyence).

**Cycloheximide-chase assay**. PAECs were treated with 20 μM/mL cycloheximide concurrent to the ActA treatment. Cell lysates were collected 0, 10, 20, 30, 40, and 50 min after the treatment with 20 ng/mL ActA or vehicle, followed by immunoblotting for BMPRII. In some experiments, cells were pretreated with 20 μM Pit-Stop, 100 nM baffilomycine A, or 100 μM leupeptin to inhibit the endocytosis and lysosomal protein degradation.

**Fig. 6 EC-specific deletion of INHBA ameliorates hypoxia-induced PH in mice. a** Representative right ventricle pulse waves and RVSP measurements in INHBA-flox and INHBA-ECKO mice under either normoxic or 3-week hypoxic (10% $O_2$) conditions ($n = 6$ biologically independent animals in each group for normoxia; $n = 7$ biologically independent animals for hypoxia flox; $n = 8$ biologically independent animals for hypoxia ECKO). **b** Fulton's index measurements in INHBA-flox and INHBA-ECKO mice ($n = 5$ biologically independent animals in each group for normoxia; $n = 8$ biologically independent animals for hypoxia flox; $n = 7$ biologically independent animals for hypoxia ECKO). **c** Representative images of hematoxylin and eosin staining and immunohistochemistry for an EC marker (vWF; in magenta color) and SMC marker (α-SMA; in green color) with DAPI in the lungs of INHBA-fox and INHBA-ECKO mice. Arrows indicate the distal PA. Bars: 50 μm. Similar results were obtained in five biologically independent samples. **d, e** Quantitation of the distal PA count per 100 alveoli (**d**) ($n = 6$ biologically independent values for normoxia flox, normoxia ECKO, and hypoxia flox; $n = 5$ biologically independent values for hypoxia ECKO) and muscularized distal PA (**e**) ($n = 4$ biologically independent values in each group for normoxia; $n = 6$ biologically independent values in each group for hypoxia) in the lungs of INHBA-flox and INHBA-ECKO mice. The number of no-muscularized (N), partially muscularized (P), and fully muscularized distal arteries was counted. **f** Quantitative PCR for INHBA mRNA expression levels in ECs isolated from the lungs of patients with iPAH and healthy control subjects ($n = 10$ biologically independent cells for control; $n = 9$ biologically independent cells for iPAH). **g** ActA concentration in the culture medium of ECs isolated from the lungs of patients with iPAH and healthy control subjects ($n = 17$ biologically independent cells for control; $n = 12$ biologically independent cells for iPAH). **h** Quantitative PCR for INHBA mRNA expression levels in ECs isolated from the lungs of patients with iPAH ($n = 3$ biologically independent cells in each group) and healthy control subjects ($n = 4$ biologically independent cells in each group) with or without hypoxia (1% $O_2$) exposure for 24 h. **i** Tube-formation assay in PAECs treated with conditioned medium derived from iPAH-ECs or control ECs in the presence or absence of Follistatin (100 ng/mL) ($n = 3$ biologically independent values in each group). Bars: 200 μm. ****$P < 0.0001$; ***$P < 0.001$; **$P < 0.01$; *$P < 0.05$; #, not significant ($P > 0.05$). Exact $P$ values are shown in the Source data file. Data are presented as the mean ± SEM. Two-sided Student's $t$-test was used to analyze the differences between the groups in the distal PA count (**d** and **e**). One-way ANOVA with Tukey's post hoc test for multiple comparisons was used to analyze the differences between each study group in the RVSP, Fulton index measurements, and tube-formation assay (**a**, **b**, and **i**). Mann–Whitney $U$ test (two-sided) was used to analyze the differences between the groups (**f**–**h**). Two-way ANOVA with Tukey's post hoc test for multiple comparisons was used to analyze the differences between each study group in the PA muscularization (**e**).

---

**Echocardiography**. Mouse transthoracic echocardiography was performed using a Siemens Acuson X300 connected to a VF13-5SP probe (Siemens) to visualize the hearts of the mice after treatment with hypoxia at one day prior to sacrifice. The echocardiographic parameters that were measured included heart rate, left ventricular end-diastolic diameter, left ventricular end-systolic diameter, aortic diameter, PAAT, and aortic velocity-time integral to calculate the cardiac output (CO). Three measurements were taken for each parameter and averaged. The ejection fraction and CO were calculated using their respective formula.

**Blood pressure measurement**. The blood pressure of the mice was measured with a tail-cuff method using BP-98A-L (Softron) in a 37 °C warmer without anesthesia. Measurements were taken at the same time as echocardiography was performed, and every five consecutive measurements were averaged. The results are presented in units of mmHg.

**Right heart catheterization**. Right heart catheterization was performed to measure the RVSP with 1.4 F Millar catheter inserted into the right jugular vein under anesthesia with 2% isoflurane immediately prior to sacrifice and at organ harvest. The RVSP was calculated from the average of five pressure waves. The results are presented in units of mmHg.

**Fulton index measurement**. After organ harvest and overnight incubation in 4% PFA at 4 °C, the hearts of mice were dissected and weighed before and after the separation of the right ventricular wall from the left ventricle and septum. The data are presented as a ratio of the right ventricle to the left ventricle + septum weight.

**Quantitative real-time PCR**. RNAs were extracted from tissues or cells using RNAiso Plus (TAKARA), and then purified using a NucleoSpin RNA Clean-Up kit (Macherey-Nagel). cDNA synthesis was then performed from ~1 μg of total RNA using a PrimeScript RT Reagent Kit with gDNA eraser (TAKARA). The amount of RNA used was always identical among samples. DNase digestion step was included in the cDNA synthesis protocol. Quantitative real-time PCR was performed using LightCycler96 (Roche Science) with FastStart SYBR Green Master (Roche Applied Science). The target gene mRNA expression levels were normalized relative to 18S levels and the results are presented in arbitrary units. We regularly performed experiments 2–3 independent times with samples of $n = 1$–2 for each group. Primers used for quantitative PCR are shown in Supplementary Table 2.

**Immunoblotting**. Antibodies used for immunoblotting are shown in Supplementary Table 1. Cells or tissues were lysed with RIPA buffer and protease and phosphatase inhibitors. The concentration was equalized before boiling in the sample buffer. The samples were then separated with 10% SDS-PAGE gels, followed by transfer onto a 0.45 μm PVDF membrane (Immobilon). After blocking with 5% BSA in TBS-T for 30 min, membranes were probed with the first antibody diluted in a blocking buffer at 4 °C overnight, followed by incubation with the appropriate secondary antibody. The signals were visualized using Amersham ECL (GE Healthcare) and detected using ChemiDoc XRS+ (BioRad). Signals were quantified and regularly normalized with β-actin expression levels and the results

are presented in arbitrary units. We regularly performed experiments 2–3 independent times with samples of $n = 1$–3 for each group.

**Analysis of BMPRII phosphorylation**. After treatment with vehicle, BMP-4, or ActA for 5 min, BMPRII-GFP-transfected PAECs were lysed with CelLytic M lysis buffer and protease and phosphatase inhibitors, followed by precleaning with IgG agarose (Sigma). Cell lysates were then incubated with anti-GFP agarose (MBL) at 4 °C overnight, and immunoprecipitated proteins were boiled at 95 °C in the sample buffer. Subsequently, immunoblotting was performed with the anti-phospho-serine/threonine antibodies (Abcam) at a 1:1000 dilution. Signals were normalized with total BMPRII expression levels and presented in arbitrary units.

**Histological staining**. Lung tissues were fixed with 4% PFA for 24 h before dehydration and paraffin embedding, followed by cutting into 3-μm sections. These sections were then stained with hematoxylin and eosin to evaluate their structural differences. The sections were otherwise stained with Elastica van Gieson staining to quantify the distal pulmonary arteries (20–50 μm).

In order to quantify the number of distal pulmonary arteries, three randomly selected images (×20 magnification) from terminal bronchioles stained with Elastica van Gieson were taken and 100 alveoli adjacent to these terminal bronchioles were counted before counting the number of distal pulmonary arteries inside the area of the counted alveoli area. Images were captured using a Keyence BZ-X800 fluorescence microscopy (Keyence). Data are presented as the number of distal pulmonary arteries per 100 alveoli.

**Immunofluorescence staining**. Antibodies used for immunostaining are shown in Supplementary Table 1. After deparaffinization, the sections were treated with an antigen unmasking solution (Vector Laboratories) prior to incubation with anti-von Willebrand factor (Abcam) and FITC-labeled anti-actin, α-smooth muscle (Sigma-Aldrich) antibodies at 4 °C overnight. Subsequently, the sections were washed and incubated with Cy3 fluorescence-labeled goat anti-rabbit secondary antibodies (Abcam), followed by mounting with VECTASHIELD mounting medium with DAPI (Vector Laboratories). Images were captured with a Keyence BZ-X800 fluorescence microscopy (Keyence). For some immunostaining experiments, images were obtained using an LSM700 laser confocal microscope (Zeiss). In order to assess distal pulmonary artery muscularization, three randomly selected fields of view (×40 magnification) were captured, and the total number of distal pulmonary arteries (diameter; <50 μm) was determined for each group. A distal pulmonary artery was considered to be partially muscularized when 25–75% of the total diameter of the artery (determined by the vWF-positive cells) was α-SMA-positive, and be considered fully muscularized when 75–100% of total artery diameter was α-SMA-positive. The muscularized distal pulmonary arteries were then counted and divided by the total number of distal pulmonary arteries in one image. Data are presented as percentages of muscularized pulmonary arteries.

**Cell culture**. Human PAECs and human SMCs (PASMCs) were purchased from PromoCell. hMVECs-L and hMVECs-D were purchased from Clonetics. PAECs were cultured in HuMedia-EG2 (Kurabo), and PASMCs were cultured in Smooth

Muscle Cell Growth Medium 2 (PromoCell). hMVECs-L and hMVECs-D were cultured in EGM-MV2 (Lonza).

**Analysis for BMPRII trafficking**. PAECs were first seeded at a density of 30,000 cells/well onto eight-well chamber slides and then cultured for 24 h, followed by infection with a BMPRII-GFP retrovirus. After changing the medium to a normal growth medium, the cells were treated with LysoTracker and incubated overnight. On the next day, the cells were subjected to treatment with vehicle, BMP-4, or ActA for 30 min, followed by fixation with 4% PFA and mounted with a VEC-TASHIELD mounting medium with 4′,6-diamidino-2-phenylindole (DAPI) (Vector Laboratories). Images were captured using an LSM700 laser confocal microscope (Zeiss).

**Animal study and generation of genetically modified mice**. All experimental study protocols were approved by the Ethics Review Committee for Animal Experiments of Kobe Pharmaceutical University (#2019-043 and #2019-044). We have complied with all relevant ethical regulations. The plasmid containing the VE-cadherin promoter was a gift from Dr. Mochizuki and Dr. Nakaoka (National Cardiovascular Research Center). VEcad-INHBA-TG (C57BL6J background) mice were propagated as heterozygous TG animals by breeding with WT mice. Littermate WT mice were always used as control mice. INHBA-floxed mice were generated using CRISPR-Cas9 system at the Laboratory Animal Resource Center of University of Tsukuba. Briefly, an *inhibin-β-a* gene targeting was carried out using homology-directed-repair induced by dual CRISPR-mediated double-strand-break in fertilized eggs delivered from C57BL/6. The following 23 bp sequences were used for INHBA CRISPR targets: 5′-GTCCCATAAGATCTCCCGGCTGG-3′; 5′-GAG TTGGGGGTAAGGTTCGCTGG-3′. The left loxP site was inserted 1.07 kb upstream of exon 4, and the right loxP was introduced 0.7 kb downstream of exon 4. For construction of targeting vector, two of 2.0 kb INHBA genomic DNA fragments were isolated from C57BL/6 mouse as 5′- and 3′-arm fragments, respectively, so that 2.27 kb INHBA genomic DNA sequence containing exon 4 was introduced as loxP-flanked region. These INHBA-flox mice were crossed with VEcad-Cre-Tg mice to generate EC-specific conditional INHBA knockout (INHBA-ECKO) mice. VEcad-Cre-Tg mice were kindly provided by Dr. Mochizuki and Dr. Nakaoka. Littermate non-Tg/INHBA-flox mice were used as control mice. Mice were housed in designated cages of sufficient size (1–3 mice in one cage) in animal facility in which the temperature (~23 °C) and humidity (~60%) are regulated appropriately. Mice were maintained under a 12-h light/12-h dark cycle, and fed normal chow (containing 23.1% protein and 5.1% fat) with ad libitum access to water and food. Mice were regularly used for experiments at 8–12 weeks of age. In order to prepare a PH model, the mice were exposed to hypoxia (10% $O_2$) for 3 weeks.

**EC isolation from the mice**. Mouse lung ECs (MLEC) isolation was performed using a magnetic sorting system. Lung specimens were processed using a mouse lung dissociation kit and a gentleMACS Dissociator (Miltenyi Biotec), followed by anti-CD146 magnetic microbeads (Miltenyi Biotec) as previously reported with minor modifications[46,47]. Briefly, lung tissues were dissociated into a single-cell suspension using the disassociation protocol as per the manufacturer's instructions, followed by magnetic sorting using anti-CD146 antibodies (Miltenyi Biotec). The resulting isolated EC suspension was then plated and cultured in HuMedia-EG2 until reaching ~90% confluency before collection. For some experiments, cells in the flow-through were used as non-ECs.

**Cultured human pulmonary endothelial cells**. We have complied with all relevant ethical regulations including the declaration of Helsinki. The study protocols were approved by the local ethics committee (CPP Ile-de-France VII, Le Kremlin-Bicêtre, France). All patients gave informed consent before the study.

Human pulmonary ECs were isolated and cultured as previously described[48–51]. A 5-cm³ lung tissue fragment was digested by incubating with Dispase I at 37 °C for overnight. The suspension was filtered, plated onto 0.1% gelatin-coated wells, and grown in MCDB131 medium supplemented with 10% FCS, 50 U/mL of penicillin/streptomycin, 4 mmol/L L-glutamine, 25 mmol/L HEPES, 10 U/mL heparin, 1 μg/mL human endothelial cell growth supplement, and 10 ng/mL vascular endothelial growth factor. The isolated ECs were strongly positive for acetylated low-density lipoprotein coupled to Alexa 488 (Alexa 488-Ac-LDL), von Willebrand factor (vWF), CD31, and for *Ulex europaeus* agglutinin-1 (UEA-1) and negative for alpha-smooth muscle actin (α-SMA). These patient-derived PAECs have not been tested for PAH-associated germ-line mutations. Cells were used between passages 3 and 6.

In order to evaluate expressions of INHBA and ActA, pulmonary ECs derived from control or iPAH patients were grown to 70–80% of confluence in T-75 culture flasks and then exposed to 0.3% FCS for 24 h. Conditioned medium was used to assess ActA secretion with a specific ELISA kit (Bio-Techne). Total RNA from the pulmonary ECs was extracted using Qiagen's RNeasy Micro Kit according to the manufacturer's guidelines and the high capacity RNA-to-cDNA kit (Applied Biosystems) was used to generate high-quality full-length cDNA. Levels of mRNA encoding INHBA were measured by real-time quantitative PCR using TaqMan gene expression assay (Applied Biosystems).

To evaluate the effects of hypoxia on IHNBA expression, human pulmonary ECs were grown to 70–80% of confluence in T-25 culture flasks and then exposed to hypoxia (0.1% $O_2$) for 24 h in presence of media containing 0.3% FCS. Conditioned medium was used to assess ActA secretion and human pulmonary ECs were collected to determine levels of IHNBA mRNA.

**Statistical analysis**. All data are presented as the mean ± standard error of the mean (SEM). Differences between two groups were analyzed using two-tailed Student's *t*-test or Mann–Whitney U test when appropriate. Differences between three or more groups were analyzed with one-way or two-way ANOVA with Tukey's post hoc test or Fisher's exact test. A *P*-value of <0.05 was considered statistically significant. All statistical analyses were performed using GraphPad Prism 8 (GraphPad Software Inc.).

**Reporting summary**. Further information on research design is available in the Nature Research Reporting Summary linked to this article.

## Data availability
The authors declare that all data supporting the findings of this study are available within the paper and its supplementary information files. The DNA microarray data have been deposited to the Gene Expression Omnibus with the dataset identifier GSE156225 and GSE156233. All remaining data will be available from the corresponding author upon reasonable request. Source data are provided with this paper.

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

## Acknowledgements
This work was supported by the JSPS KAKENHI (JP16H05827). We thank Yoko Suzuki for the excellent technical assistance.

## Author contributions
G.R.T.R. performed most of the experiments and data analyses. K.M. performed some of mouse experiments. T.F. and M.Y. generated the INHBA-flox mice. L.T., C.G., and M.H. organized and performed experiments using human samples. G.R.T.R., K.I., and N.E. wrote the manuscript. K.I., K.H., and N.E. conceived and supervised overall experiments.

## Competing interests
The authors declare no competing interests.
