## [Peer Review File · Nature Communications]

REVIEWER COMMENTS

Reviewer #2 (Remarks to the Author):

In this manuscript, Ryanto et al test the hypothesis that activin A overexpression from endothelial cells in the pulmonary vasculature promotes the development of pulmonary arterial hypertension suggest that this is reliant upon a link with BMPR-II. Their hypothesis is derived from expression array analyses of endothelial cell and human tissues that indicate enriched activin A expression in lung microvascular endothelial cells and lung tissue. They undertake in vitro studies of endothelial cell network formation and apoptosis viral overexpression of Activin A. They subsequently examine the interactions with BMPR-II. For in vivo assessment of these studies, they have generated two mouse models, the first being a mouse overexpressing activin A in endothelial cells using a VE-cadherin and the second is a mouse with an endothelial-cell specific knockout of activin A. They subject both of these models to 3 weeks of hypoxia to assess the pulmonary hypertensive phenotype, suggesting that Activin A overexpression exacerbates pulmonary hypertension, whereas knockout reduces this phenotype. Although this manuscript presents some interesting data, I have several concerns regarding the extent of phenotyping of the mouse models, the in vitro studies and the lack of information provided in the methods, which makes interpretation of the findings difficult.

Major comments:

1) The authors present two mouse models, the VEcad-INHBA-TG overexpressing mouse and the INHBA-ECKO endothelial cell specific knockout. There are several issues:

a) There are no data provided on overall phenotyping (size at birth, bodyweight)

b) There are no data relating to activin A (or related molecules such as the other inhibin genes, follistatin) expression in tissues other than lung in these models. Nor are any data provided regarding circulating activin levels.

c) No information is provided regarding the morphology of the lung vasculature during development or at birth. Do they have variation in vessel densities?

d) Are there any cardiac defects in the mice? Are their total heart weights different? (only Fulton index is provided)

e) In Figures 5d-h, 6a-e and Supplementary Figure 4b, why are data for hypoxic mice presented, but no data for the normoxic baselines? These mice should be phenotyped fully.

f) The VE-cadherin promoter used in the mice is also expressed in haematopoietic cells. As this may be important in the context of PAH, what is the expression in haematopoietic cells?

2) The experimental details in this manuscript are severely lacking and this restricts the capacity of the reader to make a proper judgement of the results presented, specifically

a) No information is provided regarding how many biological replicates of the PAECs, HCAEC, HMVEL-L, HMVEC-D and HPASMC were used in either the microarray analyses or the cell expression analyses (Figure 1 and methods).

b) For the array analysis, no information is provided regarding cell seeding and whether or not they were grown to confluence. Nor are any media conditions provided for the conditions the cells were in when they were lysed.

c) No information is provided regarding the number of biological replicates of the tissue RNA samples used for the array analysis (Figure 1 and methods).

d) There is a lack of information relating to the viral transductions. Was any titration of the viruses undertaken before they were used? What was the MOI or pfu used for transduction? Were virus samples aliquoted and stored at -80oC, or were they freeze-thawed? How many times were viral preparations generated? Polybrene is mentioned in Supplementary table 3 Was this used for transduction and if so, what concentration was used?

e) The vehicle conditions are not provided for Figure 2 or Figure 3. Are the cells in full growth media or are they in a low serum/growth factor medium? This is important as both the growth factors used for supplementation of endothelial growth media and serum factors such as TGF-beta and BMP9 can influence BMPR-II and the Smad and ID pathways.

f) What are the conditions for the cycloheximide chase experiments? The concentration and diluent for the CHX are not provided and the treatment conditions have not been described in the methods.

g) How are the tube lengths defined and measured?

h) What are the concentrations of PitStop and Bafilomycin A used in the experiments.

3) There is an absence of key controls in several experiments.

a) In Figure 2, the authors use follistatin as an activin A antagonist. They do not include follistatin alone. Nor do they include follistatin in Figure 2c to demonstrate that they are directly inhibiting activin A with the follistatin they are using.

b) In Figure 3a, 3b, 3c, 3h and 3i, there is no baseline untransfected control. Overexpression of GFP can disrupt cell behavior and without this control, the potential effect of GFP overexpression cannot be assessed.

4) In both the introduction and discussion, the capacity for BMPR-II to bind to activin A is discussed. However, the authors do not provide any context in terms of the roles of the type I receptors required for this signalling and do not address this throughout this study. This needs to be addressed in the context of what we know regarding endothelial cells.

5) The authors use cells virally overexpressing activin A for many of their experiments but there is no real validation of these cells regarding the specificity of the system. What are the measured levels of

activin A in the conditioned media from these cells? Have the authors assessed these cells for any changes in inflammatory cytokines, interferons or other members of the inhibin family produced as a consequence of this overexpression system? What about the expression of key BMP inhibitors such as Gremlin-1? Aside from BMPR-II, have the authors assessed the levels of the main BMP type I (ALK1, ALK2, ALK3) and type II receptors (ACTRIIA and ACTRIIB) to assess if these are altered as a consequence of the overexpression system? What happens to these receptors in the BMPR-II overexpression system and in the combined BMPR-II+INHBA overexpressing cells?

6) In Figures 3 and 4, the authors compare responses to BMP4 and activin A in their experiments. They do not observe particularly robust responses to BMP4, which may be due to the insensitivity of their cells to this ligand. They should assess Smad phosphorylation and/or Id1 responses in their endothelial cells to show that the BMP4 is inducing a robust signal. The only evidence provided is the slight increase of phosphorylation of overexpressed BMPR-II in Figure 4g, which is not necessarily indicative of signalling.

7) No data are presented with respect to activin A signalling through Smads. For example, in supplement figure 2, how do the authors know that the activating A treatment has worked?

8) The authors use the appearance of condensed nuclei as their definition of apoptosis. However, nuclear condensation alone is not valid as a readout of apoptosis. For this assay, the cells should be co-stained with either TUNEL reagent or co-stained for cleaved caspase-3 or cleaved caspase-7 to confirm that the cells are apoptotic. In addition, at least two complimentary methods should be used to assess apoptosis. Therefore, an additional assay, such as Annexin-V/PI staining should also be used. If there is a dramatic increase in some form of cell death, does this also mean that the tube formation assays may be indicative of dying cells rather than any angiogenic responses?

9) The authors claim at the end of the results section titled: "INHBA/ActA negatively regulates the EC function" that "These data collectively indicate that overabundance of EC-derived INHBA/ActA negatively regulates the EC function in an autocrine manner". This is not true based on the data they present. They only present data from a system where they are artificially overexpressing activin A in endothelial cells or applying exogenous activin A. They do not present any data relating to an overabundance of endogenously produced activin A.

10) The authors need to be careful with their conclusions regarding the BMPR-II localisation studies. It has been shown that overexpression of BMPR-II, like other proteins, leads to BMPR-II being diverted to the proteosomal pathway for degradation, whereas endogenous BMPR-II is degraded via the lysosomal pathway. Also, they state that the BMPR-II is localised to the cell surface, but this is not clear in the images presented. In the context of endothelial cell function, why have they used single cells rather than confluent endothelial monolayers?

11) In Figure 4g, the authors have treated cells with activin A or BMP4, followed by immunoprecipitation and probing with an anti-phospho serine/threonine antibody as a means to indicate BMPR-II activation. However, this will label all phosphorylated serines and threonines and will not discriminate between individual sites. Therefore, the authors cannot claim that the phosphorylation events mediated by BMP4 and activin A are identical and both represent activation

as they might be detecting phosphorylation events leading to positive signalling in one case and negative signalling in the other. I refer back to comment 6 above.

12) As BMPR-II reduction is the most common genetic defect underlying both heritable and idiopathic forms of PAH, the authors have not addressed the influence of activin A in the context of reduced BMPR-II. To draw firm conclusions regarding their studies, this needs to be addressed.

Minor comments

1) In the supplement, the methods state that “After centrifuged at 1,500 rpm for 5 min....”. rpm is meaningless without any indication of rotor diameters. Centrifugation information should be provided as RCF (g).

2) There is no reference to Supplementary tables 1-3 in the relevant methods sections.

3) In the description of the tube formation assay (supplement), it is stated that wells were photographed at 1 hour intervals. How was it ensured that the same field of view was imaged each hour for each well?

4) In the Quantitative real-time PCR method description, the authors state that “cDNA synthesis was then performed from approximately 1 µg of total RNA using a PrimeScript RT Reagent”. Did the amount of RNA used differ for individual samples? Was a DNase digestion step included in the RNA extraction protocol?

5) For assessment of muscularization, why were fully and partially muscularized vessels counted together? It is better practice to divide the data into non-muscularized, partially muscularized and fully muscularized.

6) In Figure 1A, the legend states “Venn Diagram analysis and heatmap of genes expressed in the MVEC-Ls, which were three times greater than in MVEC-Ds, CAECs and PAECs (upper one)”. This is confusing as it is not clear if the data are presenting the expression data and from these data, some genes were 3-times greater, or if the data presented are for genes that were three times greater. Fig1a should be divided into 4 panels and each described to make this clearer. Also, there is no definition for the red boxes.

7) In Figure 2b, the legend states that apoptosis was induced with hydrogen peroxide, yet this is not mentioned in the methods.

8) Please add molecular weight markers to Western blots.

9) It is recommended not to use red and green for immunofluorescence – please change the colors according to the current journal guidance.

10) In Figure 3i, the BMPR-II overexpressing cells seem to form thicker connections due to a higher level of cell recruitment. It would be useful to capture this in another measure.

11) The BMPR-II bands in the vehicle lane for Figure 4c are very faint, how did the authors manage to quantify these?

12) The results section pertaining to supplement figure 4 do not indicate that these data are from hypoxic mice.

13) Have the patient-derived PAECs been tested for PAH-associated germ-line mutations?

14) Does conditioned media from the mutant PAECs cause a greater inhibition of network formation than media from control cells? If so, is this inhibited by follistatin?

15) The authors need to address the relevance of these data in the context of Sotatercept.

Reviewer #3 (Remarks to the Author):

This manuscript provides evidence of a role for endothelial INHBA/ActA in pulmonary arterial hypertension through an effect on BMPRII downregulation and signalling. Key mechanistic data is provided in Fig. 4, where data are shown consistent with ActA in human pulmonary artery endothelial cells (PAECs) causing increased endocytosis and lysosomal degradation of BMPRII. Using a cycloheximide chase assay, quantitative immunoblotting with appropriate statistical analysis showed that ActA increased the degradation of BMPRII and this was prevented by the PitStop, an inhibitor of endocytosis or bafilomycinA, which the authors describe as a lysosome inhibitor. As further evidence that ActA simulates BMPRII traffic to lysosomes the authors show some fluorescence images of colocalization of GFP-tagged BMPRII with LysoTracker (Fig 4b) and images consistent with PitStop preventing this colocalization and bafilomycin strengthening the BMPRII signal colocalizing with LysoTracker. There are two major points that should be addressed with regard to these data:

1. The colocalization data needs to be quantified using Pearson's and/or Manders' correlation coefficients. On their own the images shown are not completely convincing. Other studies have used colocalization coefficients to show the overlap of a cell surface protein and a lysosomal marker under various conditions, for example GLUT1 and LAMP1 as shown in Steinberg F et al. (2013) *Nature Cell Biol.* 15:461-71. The present authors may also find LAMP1 a more appropriate marker for lysosomal/late endosomal compartments if they have any difficulties with retaining the LysoTracker staining following fixation.

2. Bafilomycin A is usually described as blocking lysosomal protein degradation through inhibition of the lysosomal V-ATPase and a consequent increase in lysosomal pH. Although the use of bafilomycin A is appropriate in the experiments reported, there have been some previous studies suggesting that its action, when added to cells, is not limited to inhibition of lysosomal V-ATPase, see e.g. Mauvezin C et al (2015) *Nature Commun* 6:7007. doi: 10.1038/ncomms8007. Therefore, the manuscript would be improved if the interpretation of the bafilomycin A experiments was backed up by use of lysosomal protease inhibitors such as leupeptin, E64 etc. Interpretation of experiments using PitStop also requires caution, see eg Willox AK et al (2014) *Biol Open* 3: 326-331, but this is a lesser issue for

the present study because the key question is whether ActA stimulates BMPRII delivery to and degradation in lysosomes.

Reviewer #4 (Remarks to the Author):

In this paper, the authors reported that excessive production of inhibin- β -A (INHBA) by endothelial cells (ECs) impairs the EC function in an autocrine manner by functioning as activin A (ActA), which induces internalization and degradation of bone morphogenetic protein type 2 (BMPRII). The presented data indicated that dysregulated EC ActA-BMPRII plays a crucial role in the progression of pulmonary artery hypertension (PAH) and can be a target for developing treatment of PAH.

The study is very well designed and executed. I only have some minor comments.

1. Have authors analyzed if INHBA expression is higher in ECs isolated from PAH patients than normal lung ECs?
2. For Fig. 2b, the authors claimed that the conditioned medium of INHBA-overexpressing PAECs tended to enhance apoptosis in PAEC. However, $P=0.09$ or 0.1 is too far from tended. The authors should consider to increase the sample size to demonstrate the significance.
3. The data showed in Supplementary Fig. 1 are not convincing.

The value of GFP group was driven by an outlier.

4. In the Results section, under the subtitle "Dysregulated activation of INHBA in ECs exacerbates pulmonary hypertension", the authors wrote: "There were no differences in the blood pressure and heart ventricular function, but a significant decrease in the pulmonary artery acceleration time (PAAT) was observed in VEcad-INHBA-Tg mice compared to littermate wild-type (WT) mice (Supplementary Fig. 4)." Please clarify if this only refer to left ventricle function. Right ventricular function was impaired based on data presented in Fig. 5a.

5. Although it was mentioned in Methods section, it will be easier for reader to follow if the authors provide information about the promoter used to induce INHBA-ECKO and if the deletion is induced in the Results section, under the subtitle: "Conditional knockout of INHBA in ECs ameliorated pulmonary hypertension."

6. It is very difficult to see apoptotic nuclei in Hoechst dye stained pictures in Figs. 2 and 3. Authors should consider including zoom in pictures or use a different apoptotic marker.

7. In Fig. 5 and 6, vWF staining is not showing well. Please add arrows and zoom in pictures to show the details.
8. For Fig. 5d and e. please include microscopic images to indicate which vessels were counted.
9. For Fig. 6c, please add arrows to indicate abnormal vessels.

RESPONSES TO THE REVIEWERS' COMMENTS

Reviewer #2 (Remarks to the Author):

Comment-1: *The authors present two mouse models, the VEcad-INHBA-TG overexpressing mouse and the INHBA-ECKO endothelial cell specific knockout. There are several issues:*

- a) There are no data provided on overall phenotyping (size at birth, bodyweight)*
- b) There are no data relating to activin A (or related molecules such as the other inhibin genes, follistatin) expression in tissues other than lung in these models. Nor are any data provided regarding circulating activin levels.*
- c) No information is provided regarding the morphology of the lung vasculature during development or at birth. Do they have variation in vessel densities?*
- d) Are there any cardiac defects in the mice? Are their total heart weights different? (only Fulton index is provided)*
- e) In Figures 5d-h, 6a-e and Supplementary Figure 4b, why are data for hypoxic mice presented, but no data for the normoxic baselines? These mice should be phenotyped fully.*
- f) The VE-cadherin promoter used in the mice is also expressed in haematopoietic cells. As this may be important in the context of PAH, what is the expression in haematopoietic cells?*

Response-1:

Thank you for the comments. We have responded to each comment as follows.

- a) We provided new data regarding the overall phenotype of INHBA-TG and INHBA-ECKO mice in the revised manuscript. Both mice are viable and fertile, and there are no significant differences between WT and INHBA-TG mice, or INHBA-flox and INHBA-ECKO mice in appearance. As shown in new Supplementary Fig. S4a and Fig. S7a, body weight was not different between WT and INHBA-TG mice, or INHBA-flox and INHBA-ECKO mice both in neonates and adults.
- b) We provided expressional data for active A and related molecules in the heart of INHBA-TG and INHBA-ECKO mice in the revised manuscript. As shown in the new Supplementary Fig. S4g and Fig. S7f, expression of INHBA, INHBB, INHA, and

follistatin in the heart was not different between WT and INHBA-TG mice, or INHBA-flox and INHBA-ECKO mice.

Also, we provided new data of circulating activin A concentration in INHBA-TG and INHBA-ECKO mice in the revised manuscript. As shown in the new Supplementary Fig. S4c, serum activin A concentration was not different between WT and INHBA-TG mice, while lung ECs isolated from TG mice produce more activin A than lung ECs of WT mice. Serum activin A concentration was also similar between INHBA-flox and INHBA-ECKO mice as shown in the new Fig. S7c.

- c) There were no apparent morphological changes in the lung vasculature, and lung vessel density was similar in neonates of INHBA-TG and INHBA-ECKO mice as compared to those in WT and INHBA-flox mice, respectively (new Supplementary Fig. S4e and Fig. S7d).
- d) As shown in the new Supplementary Fig. S4f and Fig. S7e, there are no cardiac defects in INHBA-TG and INHBA-ECKO mice. Also, total heart weights were similar between WT and INHBA-TG mice, or INHBA-flox and INHBA-ECKO mice as shown in Fig. S4f and S7e.
- e) We provided the data for normoxic baseline in INHBA-TG and INHBA-ECKO mice. As shown in the new Fig. 5b, 5d, and 5e, the number of muscularized distal PA increased, while distal PA count decreased in the lung of INHBA-TG mice even under normoxic condition. Also, BMPR-II and phospho-Smad protein expression and Id1 mRNA expression in the lung were reduced in INHBA-TG mice under normoxic condition (new Fig. 5f-h). In contrast, RVSP, Fulton index, distal PA count, and the number of muscularized distal PA were not different between INHBA-flox and INHBA-ECKO mice under normoxic condition (new Fig. 6a-e).
- f) It has been reported that VE-cadherin expression in haematopoietic cells was very little in adult mice (Blood 106(3):903-905, 2005). However, according to the comments, we newly analyzed INHBA expression in bone marrow cells, and found no difference between WT and INHBA-TG mice (Supplementary Fig. S4d).

Comments-2: *The experimental details in this manuscript are severely lacking and this restricts the capacity of the reader to make a proper judgement of the results presented, specifically*

- a) *No information is provided regarding how many biological replicates of the PAECs, HCAEC, HMVEL-L, HMVEC-D and HPASMC were used in either the microarray analyses or the cell expression analyses (Figure 1 and methods).*
- b) *For the array analysis, no information is provided regarding cell seeding and whether or not they were grown to confluence. Nor are any media conditions provided for the conditions the cells were in when they were lysed.*
- c) *No information is provided regarding the number of biological replicates of the tissue RNA samples used for the array analysis (Figure 1 and methods).*
- d) *There is a lack of information relating to the viral transductions. Was any titration of the viruses undertaken before they were used? What was the MOI or pfu used for transduction? Were virus samples aliquoted and stored at -80oC, or were they freeze-thawed? How many times were viral preparations generated? Polybrene is mentioned in Supplementary table 3. Was this used for transduction and if so, what concentration was used?*
- e) *The vehicle conditions are not provided for Figure 2 or Figure 3. Are the cells in full growth media or are they in a low serum/growth factor medium? This is important as both the growth factors used for supplementation of endothelial growth media and serum factors such as TGF-beta and BMP9 can influence BMPR-II and the Smad and ID pathways.*
- f) *What are the conditions for the cycloheximide chase experiments? The concentration and diluent for the CHX are not provided and the treatment conditions have not been described in the methods.*
- g) *How are the tube lengths defined and measured?*
- h) *What are the concentrations of PitStop and Bafilomycin A used in the experiments.*

Response-2:

Thank you for the comments. We have responded to each comment as follows.

- a) For the cell expression analysis, we used 2 replicates of the cells (n=1-2 each for each time). We have newly described about the biological replicates in the Supplemental Method of the revised manuscript.
- b) We regularly used endothelial cells at P3-6 for all the experiments. Cells were cultured in appropriate growth medium (EGM-MV2 medium for MVECs and HuMedia-EG2 for other ECs). When cells reached subconfluent (~90% confluency),

cells were lysed in TRIzol to collect RNAs. This information was added in the Supplementary Method of the revised manuscript.

- c) Each human tissue RNAs obtained from BioChain were 5-donor pooled RNAs, and 1 vial of the mixed RNAs for each tissue were used for the microarray analysis. No biological replicate was used. We have newly described about this in the Supplemental Method of the revised manuscript.
- d) We did not strictly titrate the viruses; however, we regularly determined the appropriate dilution of retroviruses by analyzing the transduction efficacy of GFP-positive control whenever viruses were newly prepared. Viruses were aliquoted and stored at -80°C after preparation, and thawed immediately before use. We tried to prepare virus stocks as many as possible for each time, but 3-4 times of viral preparation was needed. When cells were infected with retroviruses, 8 µg/mL polybrene was regularly added in the medium. We have newly described these methods in the Supplemental Method of the revised manuscript.
- e) Cells were regularly cultured in the growth medium unless otherwise mentioned. We have newly described this in the Supplemental Method of the revised manuscript.
- f) For the chase experiment, CHX are used at 20 µM/mL. We have newly described this in the Supplemental Method of the revised manuscript.
- g) Images for tube-formation were printed out, and the length of tubes was manually measured. We have newly described these methods in the Supplemental Method of the revised manuscript.
- h) PitStop and Bafilomycin A were used in the experiments at the concentration of 20 µM and 100 nM, respectively. We have newly described this information in the Supplemental Method of the revised manuscript.

Comment-3: *There is an absence of key controls in several experiments.*

a) In Figure 2, the authors use follistatin as an activin A antagonist. They do not include follistatin alone. Nor do they include follistatin in Figure 2c to demonstrate that they are directly inhibiting activin A with the follistatin they are using.

b) In Figure 3a, 3b, 3c, 3h and 3i, there is no baseline untransfected control.

Overexpression of GFP can disrupt cell behavior and without this control, the potential effect of GFP overexpression cannot be assessed.

Response-3:

Thank you for the comments.

- a) We newly analyzed the tube-formation and apoptosis in PAECs of the GFP-control group treated with follistatin. As shown in the new Fig. 2a, there was no significant effect of follistatin on endothelial functions in PAECs of the GFP-control group. Also, follistatin did not affect the tube-formation and apoptosis in untransfected PAECs (new Fig. 2b) We newly analyzed the tube-formation and apoptosis in PAECs treated with both recombinant ActA and follistatin. As shown in the new Fig. 2c, follistatin completely canceled the inhibitory effects of recombinant ActA on endothelial functions.
- b) We have newly analyzed the potential effect of GFP overexpression on endothelial functions using baseline untransfected cells or MOCK-transfected cells as a control. As shown in the new Supplementary Fig. S2a, GFP overexpression did not show any biological effects on tube-formation and apoptosis in PAECs. Also, GFP-overexpression did not affect the BMPRII protein expression in PAECs (Supplementary Fig. S2a). We have included these data in the revised manuscript.

Comment-4: *In both the introduction and discussion, the capacity for BMPR-II to bind to activin A is discussed. However, the authors do not provide any context in terms of the roles of the type I receptors required for this signalling and do not address this throughout this study. This needs to be addressed in the context of what we know regarding endothelial cells.*

Response-4:

Thank you for the comments.

We have newly described the role of the type I receptors for the signaling of the TGF-beta superfamily in the Introduction of the revised manuscript.

Comment-5: *The authors use cells virally overexpressing activin A for many of their experiments but there is no real validation of these cells regarding the specificity of the*

system. What are the measured levels of activin A in the conditioned media from these cells? Have the authors assessed these cells for any changes in inflammatory cytokines, interferons or other members of the inhibin family produced as a consequence of this overexpression system? What about the expression of key BMP inhibitors such as Gremlin-1? Aside from BMPR-II, have the authors assessed the levels of the main BMP type I (ALK1, ALK2, ALK3) and type II receptors (ACTRIIA and ACTRIIB) to assess if these are altered as a consequence of the overexpression system? What happens to these receptors in the BMPR-II overexpression system and in the combined BMPR-II+INHBA overexpressing cells?

Response-5:

Thank you for the comments.

We have newly measured the Act A concentration in the conditioned medium of ECs transfected with GFP, NHBA, BMPR-II, and INHBA+BMPR-II. As shown in the new Supplementary Fig. S2e, overexpression of INHBA caused many Act A production, whereas BMPR-II overexpression did not affect the Act A production in ECs. Also, we newly analyzed the expression of inflammatory cytokines, interferons, other members of the inhibin family, Gremlin-1, BMP type I (ALK1, ALK2, ALK3) and type II receptors (ACTRIIA and ACTRIIB) in these cells. The results were shown in the new Supplementary Fig. S2f. There was no significant effect of INHBA and/or BMPRII overexpression on most of these genes expression, while INHBA-overexpression enhanced interferon- β expression in PAECs. These results were described in the Results and Discussion of the revised manuscript.

Comment-6: *In Figures 3 and 4, the authors compare responses to BMP4 and activin A in their experiments. They do not observe particularly robust responses to BMP4, which may be due to the insensitivity of their cells to this ligand. They should assess Smad phosphorylation and/or Id1 responses in their endothelial cells to show that the BMP4 is inducing a robust signal. The only evidence provided is the slight increase of phosphorylation of overexpressed BMPR-II in Figure 4g, which is not necessarily indicative of signalling.*

Response-6:

Thank you for the comments.

We newly analyzed the Smad signaling in PAECs treated with BMP4. As shown in the new Fig. S2b, BMP4 induced robust Smads phosphorylation in PAECs, indicating that PAECs are sensitive to BMP4.

Comment-7: *No data are presented with respect to activin A signalling through Smads. For example, in supplement figure 2, how do the authors know that the activating A treatment has worked?*

Response-7:

Thank you for the comment.

We newly analyzed Act A signaling through Smads in ECs and SMCs, and confirmed that activin A treatment worked in these cells as shown in the new Supplementary Fig. S2c and S2d.

Comment-8: *The authors use the appearance of condensed nuclei as their definition of apoptosis. However, nuclear condensation alone is not valid as a readout of apoptosis. For this assay, the cells should be co-stained with either Tunel reagent or co-stained for cleaved capase-3 or cleaved caspase-7 to confirm that the cells are apoptotic. In addition, at least two complimentary methods should be used to assess apoptosis. Therefore, an additional assay, such as Annexin-V/PI staining should also be used. If there is a dramatic increase in some form of cell death, does this also mean that the tube formation assays may be indicative of dying cells rather than any angiogenic responses?*

Response-8:

Thank you for the comments.

We have newly analyzed the apoptosis using TUNEL-staining and Annexin-V/PI staining. As shown in the new Fig. 2 and Supplementary Fig. S1a-c, apoptosis

assessed by TUNEL-staining and Annexin-V/PI-staining showed results similar to the apoptosis previously assessed by the appearance of condensed nuclei.

In the apoptosis assay, we induced apoptosis by depleting serum and growth factors for 24 h. In contrast, cells were cultured in the growth medium for the tube-formation assay, and incubated for up to 8 h. Therefore, we presume that increased apoptosis play no or minimal role in the impaired tube-formation in ECs treated with INHBA/Act A.

Comment-9: *The authors claim at the end of the results section titled: “INHBA/ActA negatively regulates the EC function” that “These data collectively indicate that overabundance of EC-derived INHBA/ActA negatively regulates the EC function in an autocrine manner”. This is not true based on the data they present. They only present data from a system where they are artificially overexpressing activin A in endothelial cells or applying exogenous activin A. They do not present any data relating to an overabundance of endogenously produced activin A.*

Response-9:

Thank you for the comments.

We agree that it is not appropriate to conclude “overabundance of EC-derived INHBA/ActA negatively regulates the EC function in an autocrine manner” according to the data of gain-of-function experiments. We therefore described as “These data collectively suggest that overabundance of EC-derived INHBA/ActA could negatively regulate the EC function in an autocrine manner, though biological effects of overabundance of endogenously expressed INHBA/Act A need to be analyzed” in the revised manuscript.

Comment-10: *The authors need to be careful with their conclusions regarding the BMPR-II localisation studies. It has been shown that overexpression of BMPR-II, like other proteins, leads to BMPR-II being diverted to the proteosomal pathway for degradation, whereas endogenous BMPR-II is degraded via the lysosomal pathway. Also, they state that the BMPR-II is localised to the cell surface, but this is not clear in*

the images presented. In the context of endothelial cell function, why have they used single cells rather than confluent endothelial monolayers?

Response-10:

Thank you for the comments.

Because it is difficult to detect the endogenously expressed BMPR-II in ECs by immunocytochemistry, we overexpressed BMPRII-GFP to analyze its subcellular localisation. We newly described the limitation of this experiment because of the use of exogenously expressed BMPR-II in the Discussion of the revised manuscript.

To confirm the cell-surface expression of BMPRII-GFP, we newly labeled the plasma membrane using CellMask (Thermo). As shown in the new Supplementary Fig. S3a, BMPRII-GFP was largely co-localised with the plasma membrane, indicating its predominant localisation on the plasma membrane.

In order to improve the transfection efficacy, we used ECs at 50~60% confluency; therefore, we did not use confluent endothelial monolayers.

Coment-11: *In Figure 4g, the authors have treated cells with activin A or BMP4, followed by immunoprecipitation and probing with an anti-phospho serine/threonine antibody as a means to indicate BMPR-II activation. However, this will label all phosphorylated serines and threonines and will not discriminate between individual sites. Therefore, the authors cannot claim that the phosphorylation events mediated by BMP4 and activin A are identical and both represent activation as they might be detecting phosphorylation events leading to positive signalling in one case and negative signalling in the other. I refer back to comment 6 above.*

Response-11:

Thank you for the comments.

To analyze whether activin A could activate BMPRII as its canonical ligands do, we assessed the BMPRII phosphorylation in response to activin A.

However, we agree with the comment, and described that the results obtained do not necessarily mean that the phosphorylation events in BMPRII mediated by BMP4 and

activin A are identical, and it remains unclear whether activin A could induce intracellular signaling through BMPRII in the Results of the revised manuscript.

Coment-12: *As BMPR-II reduction is the most common genetic defect underlying both heritable and idiopathic forms of PAH, the authors have not addressed the influence of activin A in the context of reduced BMPR-II. To draw firm conclusions regarding their studies, this needs to be addressed.*

Response-12:

Thank you for the comment.

According to the comment, we discussed the potential role of Act A in the context of reduced BMPR-II in the Discussion of the revised manuscript.

Minor comments

1) *In the supplement, the methods state that “After centrifuged at 1,500 rpm for 5 min....”. rpm is meaningless without any indication of rotor diameters. Centrifugation information should be provided as RCF (g).*

Response:

We have provided RCF (g) for the centrifugation information in the Supplementary Method of the revised manuscript.

2) *There is no reference to Supplementary tables 1-3 in the relevant methods sections.*

Response:

We have described the reference to Supplementary tables 1-3 in the relevant methods sections in the revised manuscript.

3) *In the description of the tube formation assay (supplement), it is stated that wells were photographed at 1 hour intervals. How was it ensured that the same field of view was imaged each hour for each well?*

Response:

I am sorry for the unclear description.

Because the tube-forming speed is varied among experiments, we regularly take pictures every 1 h between 4-8 h after seeding cells on the Matrigel to determine the optimal time point for the assessment of tube-formation capacity. Therefore, assessment of the tube-formation capacity was performed at a single optimal time point. We have described these methods in the Supplementary Method of the revised manuscript.

- 4) *In the Quantitative real-time PCR method description, the authors state that “cDNA synthesis was then performed from approximately 1 µg of total RNA using a PrimeScript RT Reagent”. Did the amount of RNA used differ for individual samples? Was a DNase digestion step included in the RNA extraction protocol?*

Response:

The amount of RNA used for cDNA synthesis is always identical among samples. Because the volume of RNA for the cDNA synthesis reaction was determined, sometimes we used RNA less than 1 µg in case the RNA concentration was low. Even in that case, we used identical amount of RNA among all samples. DNase digestion step was included in the cDNA synthesis protocol. We described these information in the Supplementary Methods of the revised manuscript

- 5) *For assessment of muscularization, why were fully and partially muscularized vessels counted together? It is better practice to divide the data into non-muscularized, partially muscularized and fully muscularized.*

Response:

According to the comment, we have newly analyzed the number of non-muscularized, partially muscularized and fully muscularized vessels, and show the analysis data in the new Fig. 5e and 6e.

6) In Figure 1A, the legend states “Venn Diagram analysis and heatmap of genes expressed in the MVEC-Ls, which were three times greater than in MVEC-Ds, CAECs and PAECs (upper one)”. This is confusing as it is not clear if the data are presenting the expression data and from these data, some genes were 3-times greater, or if the data presented are for genes that were three times greater. Fig1a should be divided into 4 panels and each described to make this clearer. Also, there is no definition for the red boxes.

Response:

I am sorry for the unclear description. To make it clear, we divided Fig. 1a into Fig. 1a and 1b, and changed the Figure legend as follows;

a. Genes whose expression in MVECs-L was three times greater than in MVECs-D, CAECs or PAECs were separately identified through the DNA microarray analysis. Venn Diagram analysis of these genes was shown. Numbers indicate the number of genes included in the sets of the intersections. We focused on genes whose expression in MVECs-L was three times greater than in all of MVEC-Ds, CAECs and PAECs (the center intersection surrounded by red box). The heat map of these particular genes was shown.

b. Genes whose expression in the lung was five times greater than in the heart, kidney or liver were separately identified through the DNA microarray analysis. Venn Diagram analysis of these genes was shown. Numbers indicate the number of genes included in the sets of the intersections. We focused on genes whose expression in the lung was five times greater than in all of the heart, kidney and liver (the center intersection surrounded by red box). The heat map of these particular genes was shown.

7) In Figure 2b, the legend states that apoptosis was induced with hydrogen peroxide, yet this is not mentioned in the methods.

Response:

We usually induced EC apoptosis by serum and growth factor depletion; however, because the conditioned media were prepared using growth medium containing serum and growth factors, we induced apoptosis through oxidative stress using hydrogen peroxide. We described these in the Supplementary Method in the revised manuscript.

8) *Please add molecular weight markers to Western blots.*

Response:

We have added the molecular weight markers to Western blots.

9) *It is recommended not to use red and green for immunofluorescence – please change the colors according to the current journal guidance.*

Response:

We have changed red color into magenta color when shown with green color as much as possible.

10) *In Figure 3i, the BMPR-II overexpressing cells seem to form thicker connections due to a higher level of cell recruitment. It would be useful to capture this in another measure.*

Response:

Thank you for the comment.

Although it is an interesting finding that BMPR-II overexpressing cells appeared to form thicker connections, there is no reliable method to quantify the tube thickness.

Furthermore, interpretation for these findings in the context of EC angiogenic capacity has not been established as long as I know. Therefore, we did not quantify these findings for this time.

11) *The BMPR-II bands in the vehicle lane for Figure 4c are very faint, how did the authors manage to quantify these?*

Response:

Although the bands were faint, we could quantify the density of bands as long as they were visible.

12) *The results section pertaining to supplement figure 4 do not indicate that these data are from hypoxic mice.*

Response:

I am sorry for the error in the legend. The previous Supplementary Fig. 4b shows the echocardiographic data obtained from mice under normoxic condition. We have shown echocardiographic data in WT and Tg mice under both normoxic and hypoxic conditions in new Supplementary Fig. 5b in the revised manuscript.

13) *Have the patient-derived PAECs been tested for PAH-associated germ-line mutations?*

Response:

Thank you for the comment.

Patient-derived PAECs have not been tested for PAH-associated germ-line mutations.

We described this in the Method of the revised manuscript.

14) *Does conditioned media from the mutant PAECs cause a greater inhibition of network formation than media from control cells? If so, is this inhibited by follistatin?*

Response:

Thank you for the comment.

We have newly analyzed the effect of conditioned media derived from lung ECs of control or PAH patient on network formation. As shown in the new Fig. 6i, conditioned media derived from lung ECs of PAH patient significantly impaired the network formation in PAECs compared to conditioned media derived from control lung ECs. Furthermore, these inhibitory effects of the conditioned media of patient-derived ECs were canceled by the addition of follistatin (new Fig. 6i).

15) *The authors need to address the relevance of these data in the context of Sotatercept.*

Response:

We have addressed the relevance of our data in the context of Sotatercept in the Discussion of the revised manuscript.

Reviewer #3 (Remarks to the Author):

Comment-1: *The colocalization data needs to be quantified using Pearson's and/or Manders' correlation coefficients. On their own the images shown are not completely convincing. Other studies have used colocalization coefficients to show the overlap of a cell surface protein and a lysosomal marker under various conditions, for example GLUT1 and LAMP1 as shown in Steinberg F et al. (2013) Nature Cell Biol. 15:461-71. The present authors may also find LAMP1 a more appropriate marker for lysosomal/late endosomal compartments if they have any difficulties with retaining the lysotracker staining following fixation.*

Response-1:

Thank you for the comments.

According to the comments, we have quantified the colocalization of BMPR-II and LysoTracker using Pearson's correlation coefficients. As shown in the new Supplementary Fig. S3b, quantified BMPR-II colocalization with lysosome was significantly enhanced when cells were treated with Act A, while less colocalization was observed in cells treated with BMP-4. These data further support our conclusion that Act A mediates internalization and targeting of BMPR-II into lysosome, leading to accelerated BMPR-II degradation.

Comment-2: *Bafilomycin A is usually described as blocking lysosomal protein degradation through inhibition of the lysosomal V-ATPase and a consequent increase in lysosomal pH. Although the use of bafilomycin A is appropriate in the experiments reported, there have been some previous studies suggesting that its action, when added to cells, is not limited to inhibition of lysosomal V-ATPase, see e.g. Mauvezin C et al (2015) Nature Commun 6:7007. doi: 10.1038/ncomms8007. Therefore, the manuscript would be improved if the interpretation of the bafilomycin A experiments was backed up by use of lysosomal protease inhibitors such as leupeptin, E64 etc. Interpretation of experiments using PitStop also requires caution, see eg Willox AK et al (2014) Biol Open 3: 326-331, but this is a lesser issue for the present study because the*

key question is whether ActA stimulates BMPRII delivery to and degradation in lysosomes.

Response-2:

Thank you for the comments.

According to the comments, we newly used lysosomal protease inhibitor leupeptin to inhibit the lysosomal protein degradation, and assessed the lysosomal degradation of BMPR-II in ECs treated with Act A. As shown in new Supplementary Fig. S3c, Act A-mediated BMPR-II degradation was canceled by leupeptin in a way similar to bafilomycin A.

We also described the possible non-specific action of PitStop in the Discussion of the revised manuscript.

Reviewer #4 (Remarks to the Author):

Comment-1: *Have authors analyzed if INHBA expression is higher in ECs isolated from PAH patients than normal lung ECs?*

Response-1:

Thank you for the comment.

As shown in Fig. 6f, INHBA expression was higher in ECs isolated from PAH patients than in normal lung ECs.

Comment-2: *For Fig. 2b, the authors claimed that the conditioned medium of INHBA-overexpressing PAECs tended to enhance apoptosis in PAEC. However, $P=0.09$ or 0.1 is too far from tended. The authors should consider to increase the sample size to demonstrate the significance.*

Response-2:

Thank you for the comment.

According to the comment, we have increased the sample size, and confirmed that conditioned medium derived from INHBA-overexpressing PAECs enhanced apoptosis in PAECs (new Fig. 2b). We also confirmed that this effect of the conditioned medium was canceled by the addition of follistatin (new Fig. 2b).

Comment-3: *The data showed in Supplementary Fig. 1 are not convincing. The value of GFP group was driven by an outlier.*

Response-3:

Thank you for the comment.

We have performed experiments using newly prepared samples, and confirmed that BMPR-II mRNA expression was not changed in PAECs overexpressing INHBA as shown in new Supplementary Fig. 1d.

Comment-4: *In the Results section, under the subtitle “Dysregulated activation of INHBA in ECs exacerbates pulmonary hypertension”, the authors wrote: “There were no differences in the blood pressure and heart ventricular function, but a significant decrease in the pulmonary artery acceleration time (PAAT) was observed in VEcad-INHBA-Tg mice compared to littermate wild-type (WT) mice (Supplementary Fig. 4).” Please clarify if this only refer to left ventricle function. Right ventricular function was impaired based on data presented in Fig. 5a.*

Response-4:

Thank you for the comments.

According to the comments, we described as “There were no differences in the blood pressure, heart rate, and heart left ventricular function between VEcad-INHBA-Tg and littermate wild-type (WT) mice (Supplementary Figs. 5a and 5b). In contrast, a significant decrease in the pulmonary artery acceleration time (PAAT) was observed in VEcad-INHBA-Tg mice compared to WT mice (Supplementary Fig. 5b), suggesting an increase in pulmonary artery pressure in VEcad-INHBA Tg mice.” in the Results section of the revised manuscript.

Comment-5: *Although it was mentioned in Methods section, it will be easier for reader to follow if the authors provide information about the promoter used to induce INHBA-ECKO and if the deletion is induced in the Results section, under the subtitle: “Conditional knockout of INHBA in ECs ameliorated pulmonary hypertension.”*

Response-5:

Thank you for the comment.

According to the comment, we have provided information about the promoter used to induce INHBA-ECKO in the Results section of the revised manuscript.

Comment-6: *It is very difficult to see apoptotic nuclei in Hoechst dye stained pictures in Figs. 2 and 3. Authors should consider including zoom in pictures or use a different apoptotic marker.*

Response-6:

Thank you for the comment.

According to the comments, we have newly assessed the apoptosis using TUNEL-staining and Annexin V/PI-staining. As shown in the new Figs.2a, 2b, 3i and Supplementary Fig. 1a-c, INHBA/Act A enhanced EC apoptosis assessed by TUNEL-staining and Annexin V/PI-staining in a way similar to that previously assessed by Hoechst nuclear staining.

Comment-7: *In Fig. 5 and 6, vWF staining is not showing well. Please add arrows and zoom in pictures to show the details.*

Response-7:

Thank you for the comment.

According to the comment, we have zoomed in pictures and added arrows to show vWF-positive endothelial cells in Fig. 5c and 6c in the revised manuscript.

Comment-8: *For Fig. 5d and e. please include microscopic images to indicate which vessels were counted.*

Response-8:

Thank you for the comment.

According to the comment, we have included microscopic images to indicate vessels that were counted in new Supplementary Fig. 6a in the revised manuscript.

Comment-9: *For Fig. 6c, please add arrows to indicate abnormal vessels.*

Response-9:

Thank you for the comment.

According to the comment, we have added arrows to indicate abnormal vessels in new Fig. 6c in the revised manuscript.

REVIEWER COMMENTS

Reviewer #2 (Remarks to the Author):

I thank the authors for having addressed my comments from the previous review. However, there are some outstanding points that need to be addressed:

1) The authors have added more detail in the Methods supplement with respect to the tube formation assay. However, there are some key details still missing.

i) The authors do not state how they define a tube. Are the lengths measured between junctions?

ii) Were the tube length measurements undertaken by a researcher blinded to the treatments?

iii) There are 3 data points on each of the graphs the graphs in Figures 2a-c, 6i and Supplementary Figure 2a. Presumably each point is the mean of several tube measurements within a single experiment? The authors should state how many measurements were made per point.

2) In the reporting summary, the authors state that randomization was used "whenever possible". In reality, this was only with respect to random images being collected for measurement of remodelling in tissue sections. There is no indication of other any other cases where randomization was used so I suggest this is clarified in the reporting summary.

3) There are a couple of spelling errors:

Manuscript line 203: "dose" should be "does"

Supplement figure 2c legend: "GAPHD" should be "GAPDH"

Reviewer #3 (Remarks to the Author):

No further comments on the revised manuscript.

Reviewer #4 (Remarks to the Author):

The authors have addressed my questions adequately. I have no more concern.

RESPONSES TO THE REVIEWERS' COMMENTS

Reviewer #2

I thank the authors for having addressed my comments from the previous review. However, there are some outstanding points that need to be addressed:

Comment-1: *The authors have added more detail in the Methods supplement with respect to the tube formation assay. However, there are some key details still missing.*

i) The authors do not state how they define a tube. Are the lengths measured between junctions?

ii) Were the tube length measurements undertaken by a researcher blinded to the treatments?

iii) There are 3 data points on each of the graphs the graphs in Figures 2a-c, 6i and Supplementary Figure 2a. Presumably each point is the mean of several tube measurements within a single experiment? The authors should state how many measurements were made per point.

Response-1:

Thank you for the comments. As the reviewer said, we have measured the lengths between branching junctions, and the sum of the lengths was considered as the tube-length. We have shown the example how we measured the tube-length below.

As we mentioned in the Reporting Summary, investigators were not blinded to sample allocation during experiments and outcome assessment. Therefore, the tube length measurements were undertaken by a researcher not blinded to the treatments.

To quantify the tube-lengths, images in low magnification (x4) were captured for each well. The lengths of tubes between the branching junctions were measured at 10 independent areas for 1 image, and averaged. Therefore, each point is the mean of 10 tube-measurements within a single experiment. For each group, 3 independent wells of cells were analyzed. These details were described in the Supplementary Method of the revised manuscript.

Comment-2: *In the reporting summary, the authors state that randomization was used "whenever possible". In reality, this was only with respect to random images being collected for measurement of remodelling in tissue sections. There is no indication of other any other cases where randomization was used so I suggest this is clarified in the reporting summary.*

Response-2:

Thank you for the comments. In addition to randomly choosing of the ROI for histological analysis, we used randomization for allocation of animals and cells. Specifically, we randomly determined the allocation of animals (normoxia or hypoxia) and allocation of cells (GFP-transfection or INHBA-transfection; vehicle group or Activin A group; vehicle group or Follistatin group; vehicle group or BMP-4 group or Activin A group; vehicle group or Bafilomycin group; vehicle group or PitStop group; etc.). We have added this information in the revised Reporting Summary.

Comment-3:

There are a couple of spelling errors:

Manuscript line 203: "dose" should be "does"

Supplement figure 2c legend: "GAPHD" should be "GAPDH"

Response-3

I am sorry for these careless mistakes. We have corrected these spelling errors, and carefully prepared the revised manuscript to avoid such errors.

Reviewer #3:

No further comments on the revised manuscript.

Response:

We appreciate the Reviewer's positive evaluation.

Reviewer #4:

The authors have addressed my questions adequately. I have no more concern.

Response:

We appreciate the Reviewer's positive evaluation.

REVIEWER COMMENTS

Reviewer #2 (Remarks to the Author):

I thank the authors for their responses. I have three comments relating to the metrics for the tube formation assay.

1) The description in the revised method supplement is not clear, especially when compared to the description provided in the rebuttal letter. It would be useful if the method as described in the rebuttal response, including the image of the measurement process, is included in the method supplement along with the information required for the clarifications for points (2) and (3) below.

2) The authors select 10 of the 16 rectangles for measurement of tube lengths. Are the same 10 rectangles selected the same for each image within an individual experiment? For example, comparing GFP vs INHBA vs BMPR2 vs BMPRII+INHBA in Figure 3i?

3) It is not clear whether the tube lengths measured in a single rectangle in the image in the rebuttal (i.e rectangle 1) were summed or averaged. It is clear that the values from each rectangle (1-10) were averaged.

Responses to the Reviewer's comments

Reviewer #2 (Remarks to the Author):

Comment

I thank the authors for their responses. I have three comments relating to the metrics for the tube formation assay.

- 1) The description in the revised method supplement is not clear, especially when compared to the description provided in the rebuttal letter. It would be useful if the method as described in the rebuttal response, including the image of the measurement process, is included in the method supplement along with the information required for the clarifications for points (2) and (3) below.*
- 2) The authors select 10 of the 16 rectangles for measurement of tube lengths. Are the same 10 rectangles selected the same for each image within an individual experiment? For example, comparing GFP vs INHBA vs BMPRII vs BMPRII+INHBA in Figure 3i?*
- 3) It is not clear whether the tube lengths measured in a single rectangle in the image in the rebuttal (i.e rectangle 1) were summed or averaged. It is clear that the values from each rectangle (1-10) were averaged.*

Responses

Thank you for the comments.

- 1) According to the comments, we clearly described the detailed method for tube-formation assay as provided in the previous "Responses to the Reviewer's comments", including the image showing a measurement example in the Supplementary Fig. 9 in the revised manuscript.
- 2) We have randomly chosen 10 independent rectangles among 16 for measurement of tube length. Therefore, rectangles chosen for measurement are different for each image within an individual experiment.
- 3) To determine the tube length, the lengths of tubes between the branching junctions were summed in a single rectangle in the image. Measurements were performed in 10 randomly chosen rectangles, and then values from each rectangle were averaged, as was mentioned by the Reviewer.

These detailed methodologies were clearly described in the Supplementary Methods of the revised manuscript.